# Nanochitin and Nanochitosan in Pharmaceutical Applications: Innovations, Applications, and Future Perspective

**DOI:** 10.3390/pharmaceutics17050576

**Published:** 2025-04-27

**Authors:** José Roberto Vega-Baudrit, Mary Lopretti, Gabriela Montes de Oca, Melissa Camacho, Diego Batista, Yendry Corrales, Andrea Araya, Badr Bahloul, Yohann Corvis, Luis Castillo-Henríquez

**Affiliations:** 1Lanotec Cenat Conare, San José 10101, Costa Rica; dbatista@cenat.ac.cr (D.B.); ycorrales@cenat.ac.cr (Y.C.); aaraya@cenat.ac.cr (A.A.); luis.castillohenriquez@ucr.ac.cr (L.C.-H.); 2Chemistry School, Universidad Nacional, Heredia 40101, Costa Rica; 3Laboratorio de Técnicas Nucleares Aplicadas a la Bioquímica y Biotecnología, Centro de Investigaciones Nucleares, Facultad de Ciencias, Universidad de la República, Montevideo 10129, Uruguay; mlopretti@gmail.com; 4CEDS, Universidad Técnica Nacional UTN, Alajuela 20101, Costa Rica; mmontesdeoca@utn.ac.cr (G.M.d.O.); kmce08@gmail.com (M.C.); 5Drug Development Laboratory LR12ES09, Faculty of Pharmacy, Monastir University, Monastir 5060, Tunisia; badrpharm07@gmail.com; 6Chemical and Biological Technologies for Health Group (UTCBS), Universite Paris Cite, 75023 Paris, France; yohann.corvis@u-paris.fr; 7Laboratory of Pharmaceutical Physical Chemistry, Faculty of Pharmacy, University of Costa Rica (UCR), San José 11501-2060, Costa Rica

**Keywords:** chitin, nanochitin, nanochitosan, chitosan, biocompatibility, drug delivery, therapeutic needs

## Abstract

Nanochitin is a nanoscale form of chitin—a polysaccharide found in the exoskeletons of crustaceans, insects, and some fungal cell walls—that is newly garnering significant attention in the pharmaceutical space. Its good properties, such as biocompatibility, biodegradability, and an easily adjustable surface, render it attractive for various medical and pharmaceutical applications. Nanochitin, from drug delivery systems and wound-care formulations to vaccine adjuvants and antimicrobial strategies, has demonstrated its strong potential in meeting diverse therapeutic needs. This review covers the background of nanochitin, including methods for its extraction and refining and its principal physicochemical and biological properties. It further discusses various hydrolysis and enzymatic approaches for the structural and functional characterization of nanochitin and highlights some pharmaceutical applications where this biopolymer has been studied. The review also addresses toxicity issues, regulatory matters, and challenges in large-scale industrial production. Finally, it underscores novel avenues of investigation and future opportunities, emphasizing the urgent requirement for standardized production methods, rigorous safety assessment, and interdisciplinary partnerships to maximize nanochitin’s potential in pharmaceutical research, demonstrating the importance of chitin in drug delivery.

## 1. Introduction

Interest in biopolymers for pharmaceutical and biomedical applications has recently surged, with an increasing focus on chitin and its derivatives [1,2]. Nanochitin, derived from chitin, is a promising biopolymer due to its abundance, sustainability, and unique properties, such as toughness, biodegradability, and high reactivity. It is increasingly being explored for various applications in sustainable and advanced manufacturing, biomedical fields, and environmental solutions; as the second most abundant polysaccharide in nature, chitin is primarily sourced from marine crustaceans, insects, and fungal cell walls [3,4]. The partial or complete deacetylation of chitin produces chitosan—Figure 1, a polycationic biopolymer extensively studied for its biomedical benefits [5]. Nanochitin can form hierarchical structures, ranging from nanofibrils to nanorods, contributing to its mechanical strength and application versatility [6,7]. Although the primary focus of this review is on the nanoscale derivatives of chitin and chitosan, such as nanochitin and nanochitosan, it is essential to provide background on the physicochemical properties and solubility behavior of their parent biopolymers. These characteristics determine the efficiency of nanoscale transformation processes and strongly influence their performance in pharmaceutical applications.

Beyond these well-researched forms, recent advancements in nanotechnology have developed nanochitin, generally referring to chitin fibrils, whiskers, or particles with at least one dimension measuring less than 100 nm [6,7,8].

This is due to the large surface area, mechanical strength, and reactive functional groups of nanochitin compared to the micro or macroscale. These properties are helpful for pharmacy-related applications, including drug delivery, tissue engineering, wound healing, and vaccine formulation [9,10]. Moreover, nanochitin is also biocompatible and biodegradable, and it has demonstrated low toxicity [11], making it a good candidate for the co-delivery of sortable pharmaceutical platforms. Its amphiphilicity is a bolstering agent in environmental applications, such as water purification [7,8,10].

This review highlights recent developments in the use of nanochitin, particularly in pharmaceutical applications. We reviewed and discussed the sources and extraction techniques of nanochitin, the physicochemical modifications that promote its functionality, and key characterization techniques that enable formulation and quality control. In addition, we explore its use in drug delivery, wound healing, immunotherapy, etc. Finally, we consider the regulatory prerequisites, toxicological aspects, and relevant difficulties preventing industrially implemented pharmacy using nanochitin [6,8].

## 2. Chitin: A Potential Precursor for Nanochitin

### 2.1. Sources of Chitin

#### 2.1.1. Insects

Insect waste-derived chitin is an alternative resource for food security and biomedicine (Figure 2). This new emerging alternative source has potentially raised low allergenic concerns [12,13,14] and is also being utilized in agriculture and bioplastic production [15].

Chitin is abundant in insect farming’s residual streams, especially in molting skins, and both LC-ECD and LC-MS/MS methods are reliable methods to measure glucosamine content [16]. Mealworm cuticles can also be a potential novel source of chitin and chitosan, as they have a global yield of 31.9% and an efficiency of 85% with enzymatic deproteinization [17].

#### 2.1.2. Fungi Species

Fungal cell walls also provide a vegetarian/vegan-friendly (based on glycoforms) route, particularly appealing in pharmaceutical and biomedical fields where access to animal sources can be restricted [18]. Present in the cell walls of fungi, they play a key role in cell stability and interactions with the environment, making them particularly interesting for biomedical applications, such as targeted drug delivery or vaccine development (Figure 2) [19].

In fungi, the regulation of chitin synthesis plays an essential role in maintaining the cell wall integrity and modulating host immune responses to fungal pathogens [20]. For this reason, fungal chitin recognition by the human immune system may help limit inflammation during fungal infections and help return immunity to a balanced state following pathogen clearance [21]. Disruption of gut fungal microbiota composition is correlated with mucosal inflammation and disease activity in patients with Crohn’s disease [22]. Intestinal microbiota dysbiosis was found in Chinese Crohn’s disease patients, manifesting high levels of pathogenic bacteria and low levels of beneficial bacterial species [23]. Table 1 shows some species that produce chitin.

#### 2.1.3. Crustacean Shells

The most popular plankton source for chitin is crustacean shells [4,28,29,30,31,32]. Chitin and its derivative, chitosan, can be recovered from shrimp shell waste (Figure 2) by chemical methods and may yield valuable byproducts for biomedical applications while reducing pollution [33].

Compared to commercial biopolymers, chitosan and chitin extracted from shrimp shells have good antibacterial activity against Gram-negative bacteria (*Escherichia coli*) [34]. Although waste from shrimp processing can generate commercially valuable biomolecules, including oil enriched in astaxanthin, protein, chitin, and chitosan, biorefineries of shrimp processing waste have not yet been extensively developed [35,36]. Ultrasound also helps produce biopolymers from shrimp shells, reduce protein content and particle size, and preserve beef [37].

Chitosan from fish scales, shrimp, and crab shells also had a higher fat-binding and water-binding capacity than commercial chitosan but lower thermal stability and deacetylation [38]. Artificial neural networks can help establish the best conditions for extracting chitosan from crabshell waste, allowing it to achieve a higher deacetylation degree and improved chitosan properties [39]. The extraction and conversion of seafood waste into valuable commercial products can be carried out, contributing to environmental protection and good human health [40].

#### 2.1.4. Squid and Snail

Chitin derived from squid pens and snail shells represent valuable alternative sources. Squid-derived chitin, primarily β-chitin, exhibits higher solubility and reactivity than α-chitin from crustaceans, favoring applications in nanomedicine. Snail shells offer a terrestrial biomass resource rich in calcium carbonate and chitin, with demonstrated use in scaffolds and antimicrobial films. Though less utilized industrially, these sources are gaining attention due to their distinct physicochemical properties [10,28].

### 2.2. Chemical Structures of Chitin and Chitosan

Chitin is a linear polysaccharide consisting of mainly N-acetyl-D-glucosamine (GlcNAc) units with β-1.4 glycosidic bonds (Figure 1). Chitosan, with amino (-NH2) and hydroxyl (-OH) functional groups in its chemical structure, makes it a highly reactive and versatile biomaterial that can be targeted to different applications [2,3]. Hydrogen bonding is one of the strongest intermolecular bonds, leading to high crystallinity and mechanical strength [1]. D-glucosamine (GlcN) units interspersed in chitosan exclude chitin, which is supposed to hold more unbound amine sets [41,42].

The presence of two amide groups and extensive intermolecular hydrogen bonding in chitin gives it an orthorhombic structure and reduces its ability to swell while immersed in water [43]. Using lactic acid for shrimp waste lactic acid fermentation associated with the freeze–pump–thaw (FPT) process successfully generated high-quality chitin and chitosan, presenting new applications for this material [44]. While chitin and its derivatives hold promise in tissue engineering, drug delivery, diagnosis, molecular imaging, antimicrobial activity, and wound healing, potential limitations and prospects exist [45].

Chitin and chitosan are eco-friendly biomaterials with distinctive qualities, such as antimicrobial properties, and they are used in a wide range of medical and non-medical applications [46,47]. Shell biorefinery can recycle waste shells to prepare high-value chemicals and materials, bringing potential environmental and economic benefits [48].

Chitosan (Figure 1) is a linear polymeric structure obtained from a chitin molecule containing the repeating unit of β-[1→4]-linked 2-amino-2-deoxy-D-glucose (deacetylated unit) and 2-acetamido-2-deoxy-D-glucose (N-acetylated unit). The presence of amino (-NH_2_) and hydroxyl (-OH) groups in its chemical structure imparts specific properties and reactivity. Unlike chitin, chitosan has amine groups in D-glucosamine units. In acidic solutions, these groups undergo protonation to form the corresponding (i.e., NH_3_⁺ ion), making chitosan positively charged and soluble in water at low pH values. Hydroxyl groups on the glucosamine units can bond hydrogen, facilitating their intermolecular interactions to form a gel or film. A higher deacetylation degree (DD) (>50%) promotes solubility in acidic solutions and increases its cationic behavior, both of which are important for biological and chemical applications. Due to protonated amino groups, Chitosan is a cationic polymer, allowing it to interact with negatively charged molecules, like proteins, lipids, and DNA. This property is exploited in drug delivery, gene therapy, and wastewater treatment [1,49,50].

Chitosan’s unique chemical structure is characterized by its linear polycation structure of high charge density, reactive hydroxyl, amino groups, and hydrogen bonding. It is also good in biocompatibility, physical stability, and processability. It can potentially be an antimicrobial agent in food and pharmaceutical preparations, but knowledge of its antimicrobial activity is required to maximize preparations and enhance their activity [51]. Table 2 summarizes a solubility profile [1]. Chitin and chitosan exhibit strong hydrogen bonding and high crystallinity, especially in α-chitin, making them insoluble in most solvents. However, solvents, such as ionic liquids (ILs) and deep eutectic solvents (DES), can disrupt these interactions [31]:ILs, like 1-ethyl-3-methylimidazolium acetate ([EMIM][OAc]) and 1-butyl-3-methylimidazolium chloride ([BMIM]Cl), dissolve chitin/chitosan by disrupting the intermolecular hydrogen bonding network, particularly between –OH and –NH_2_ groups, via ion-dipole interactions [6].DES, such as choline chloride:lactic acid or choline chloride:urea mixtures, exhibit similar mechanisms through strong hydrogen bonding with the polymer chains [6].

Some applications post-solubilization include electrospun fibers, hydrogels, and nanoparticle synthesis, with improved dispersibility and functionalization for pharmaceutical uses. Hexafluoroisopropanol (HFIP), though effective in dissolving chitosan, is highly volatile and toxic. It is only used in research-scale applications, such as electrospinning or NMR sample preparation. It requires stringent handling protocols and is not recommended for pharmaceutical formulations [11]. Additional acidic solvents, such as lactic acid, formic acid, and citric acid, promote protonation of amino groups, enhancing chitosan solubility [1,6,12,17,18,44]. EDTA acts as a chelating agent that removes metal ions stabilizing chitin/chitosan networks, facilitating structural loosening and partial solubilization [32,33].

In recent years, various structural modifications and nanostructured chitosan variants have shown great potential for biomedical applications, including tissue engineering, drug delivery, wound healing, and gene therapy.

Table 3 shows a particular product of modified chitosan, its properties, and typical applications, such as chitosan sulfate, thiolate chitosan, and grafted chitosan [1,49,54].

### 2.3. Rationale for Nanoscale Forms

Chitin and chitosan nanosizing (nanoparticles, nanofibers, or nanocrystals) greatly enhances their surface-to-volume ratio, further developing their physical, chemical, and biological properties. Table 4 describes the main improvements and references supporting them.

The properties of chitin and chitosan are markedly improved by nanosizing, which finds applications in nano-biomedicine, environmental science, and materials engineering. This surface-to-volume ratio enhances mechanical strength, colloidal stability, biocompatibility, and drug delivery efficiency. These modifications highlight nanochitin’s potential for advanced pharmaceutical formulations, including targeted delivery of pharmaceutical drugs and advanced wound dressing [12,48,55].

Due to its biocompatibility and functional groups, chitosan can be utilized to create various multifunctional nanoplatforms. They can be designed to encapsulate therapeutic agents and target moieties. This functionality enables them to act as drug delivery systems, improving efficacy on target cells while reducing toxicity to healthy cells, particularly in the case of cancer cells [56,57,58]. Control of drug release improves biodistribution due to chitosan-based nanoparticles and nanocomposites. It is particularly beneficial in treating cancer, making therapeutic delivery more efficacious [57,58].

Packed with nanochitosan membranes of fibrous membranes of polycaprolactone and water, chitosan was carried to enhance their hydrophilicity, degradation, and sustained antibacterial action. These features identify them as potential candidates for different types of applications, such as wound dressings or transdermal patches that can release drugs in response to pH under different circumstances [59].

## 3. Methods of Extraction and Fabrication of Nanochitin and Nanochitosan

Figure 3 presents a comprehensive overview of the various methodologies employed in extracting and producing nanochitin.

### 3.1. Extraction Methods for Nanochitin

Traditional methods of extracting chitin from crustacean shells involve chemical and biological processes. Recent developments include using ionic liquids and deep eutectic solvents to be more sustainable and reduce waste [28,60].

A recent method based on the acid-assisted glycerol swelling combined with colloid milling effectively ruptured hydrogen bonds of chitin, leading to a high yield and good dispersibility of nanochitin [61].

The microwave irradiation method uses irradiation to extract nanochitin using optimal conditions to reduce the time and acid concentration without significantly affecting the yield and thermal stability [41].

### 3.2. Top-Down Methods

#### 3.2.1. Acid Hydrolysis

Acid hydrolysis is one of the most common methods for preparing nanochitin and nanochitosan from chitin and chitosan. Acidic hydrolysis, i.e., hydrochloric or sulfuric acid, preferentially degrades amorphous domains in chitin, resulting in the recovery of crystalline nanoscale fibers or whiskers. You have worked on optimizing the acid concentration, temperature, and time to optimize the hydrolysis process and get the desired product traits. An example is nanochitin produced using 0.64% *w*/*w* sulfuric acid for a 2-h milling process [61,62,63].

Although acid hydrolysis works well, strong acids cause environmental problems. Combining hydrolysis methods, such as gamma radiation or microwave power, can increase efficiency; however, this might hurt the price. Another way to degrade cellulose is via enzymatic hydrolysis, which is more environmentally friendly but suffers from cost and processing time limitations [64].

#### 3.2.2. Mechanical Disintegration

Mechanical treatments, such as high pressure, ultrasonication, or ball milling, carve chitin into nanoscale fragments [48,65].

Nanochitin was produced through acid-assisted glycerol swelling and low-intensity colloid milling. This process uses glycerol and a small volume of sulfuric acid to cleave hydrogen bonds between chitin fibrils, enabling their further disassembly into nanoscale fibrils through mechanical disintegration. Under the best conditions, i.e., using 0.64% *w*/*w* sulfuric acid and 2 h of milling, nanochitin with <50 nm fibril width could be produced with approximately 85% yield [61].

Another method consists of the surface esterification of chitin by maleic anhydride, which improves the mechanical disintegration process. The diameter of the fibers is 10 nm, and they are uniform nanofibers. The esterification process takes place on the surface of chitin. It does not modify its crystallinity, leading to well-dispersed nanofibers in essential water because of the carboxylate salts on their surface [66].

In high-intensity ultrasonication, acoustic cavitation occurs, resulting in the fragmentation of large chitin flakes into nanofibers. High-intensity ultrasonication severely disrupts the structure of chitosan nanoparticles, which may affect their drug-delivery properties [67]. A 45 kHz frequency with a 20-min ultrasonic treatment produced small, high-yield chitin nanofibers, which were better than those obtained from crab shells [68]. An approach to the large-scale production of bionanofibers was proposed, and the ultrasonic method was introduced to facilitate bionanofiber extraction from natural sources, environmentally friendly, with a low tricalcium phosphate (TCP) consumption, and versatile [69].

### 3.3. Bottom-Up Methods

#### 3.3.1. Self-Assembly

Due to the great need for sustainable, biocompatible, and functional materials, the self-assembly of nanochitin and nanochitosan is an emerging research sector. Chitin and chitosan are biopolymers with unique characteristics. Chitin oligomers can undergo self-assembly under controlled conditions to generate supramolecular structures at the nanoscale [6,70,71].

Both nanochitin and nanochitosan are under study to develop green, sustainable materials for biomedical applications (e.g., immune scaffolds and medical scaffolds) and environmental applications (e.g., sustainable food packaging [7]. Through self-assembly, amphiphilic chitosan nanomicelles can significantly improve quercetin’s water solubility and biocompatibility and can be applied to food, pharmaceuticals, and cosmetics [72].

##### Nanochitosan

Hydrophobic–Hydrophilic Balance: The ability of chitosan nanoparticles to self-assemble can be customized by adding hydrophobic portions to the gadolinium molecules. This balance helps to produce micelles that can encapsulate lipophilic therapeutic drugs, which could further improve drug delivery via non-invasive oral, nasal, pulmonary, and ocular routes [71].Polyelectrolyte Complexes: In another method, complexes of polyelectrolytes are formed with polyanions, which induces the self-assembly of the chitosan nanoparticles. Recent developments highlight these interpolyelectrolyte complexes (IPECs) consisting of fully biodegradable components, which significantly enhance biocompatibility and environmental sustainability. Applications of biodegradable IPECs include controlled drug release systems, targeted tissue engineering scaffolds, and innovative biomedical formulations. Notable advantages encompass reduced environmental impact, improved biocompatibility, and enhanced biodegradability, making them suitable for medical and pharmaceutical applications. Nevertheless, these biodegradable IPECs face certain challenges, such as limited stability under specific physiological conditions, potential rapid degradation rates, and difficulties related to scalability and reproducibility in industrial production processes. Addressing these limitations is crucial for broader clinical and commercial applications. This method significantly influences drug delivery systems by enhancing biodistribution while minimizing pharmacological toxicity [71].

##### Nanochitin

Hierarchical Assemblies: Nanochitin can build hierarchical structures from the nano to macro scales. These assemblies increase the toughness and resistance of the material, making them appropriate for multi-component materials [6].Multiscale Interactions: The nanochitin’s native architecture enables multiscale interactions, which result in dynamic and functional structures. This aspect is important for delivering advanced materials that are more tunable and multipurpose [6].Self-assembly is influenced by pH, solvent polarity, and ionic strength [73,74,75]. Varying pH produces the unique morphologies discovered in chitosan-sodium alginate PECs. Fibrous structures develop at low pH (3 to 7), while at high pH (approximately eight and higher), colloidal nanoparticles are produced [74].

Also, pH influences self-assembly, impacting the morphology and charge of the resulting structures. Although solvent polarity is not directly addressed, it is known to affect biopolymer interactions. Ionic strength is essential for controlling the shape and structure of self-assembled complexes, presenting potential applications in biomaterials and drug delivery systems.

Figure 4 presents micelle formation, polyelectrolyte complex (PEC) formation, and nanofibril assembly under varying pH and ionic strength conditions, corresponding to mechanisms that are critical for drug delivery and biomedical applications.

A closely aligned example is presented in Montroni et al. (2019) [73], which shows the self-assembly behavior of β-chitin nanofibrils in aqueous environments. Their graphical data parallels the processes shown in our schematic, especially regarding fibrillar formation and structural rearrangements driven by environmental triggers. Additional relevant literature includes Quiñones et al. (2018), who discuss pH-dependent formation of chitosan nanoparticles via self-assembly, and Wang et al. (2019) [72], who elaborate on the amphiphilic properties of chitosan that lead to micelle formation [71,73].

#### 3.3.2. Biosynthetic Approaches: Fungi and Microbial

Fungi and selected microbial cultures can produce chitin de novo, sometimes in nanostructured forms [18]. Chitosan can be isolated from different types of fungi (*Benjaminiella poitrasii, Agaricus bisporus*, and *Pleurotus sajor-caju*). These fungi also supply high deacetylation of chitosan, which is essential for its functional properties [76]. The preparation process for producing nanochitin from fungal residues (*Hericium erinaceus*) involves mineral/protein purification and TEMPO-mediated oxidation to prepare nanochitin from fungal residues, like *Hericium erinaceus*. Fungal nanochitin prepared from *Hericium erinaceus* residue shows improved structure, dispersity, and gelation ability, making it a promising biocompatible material for various applications [77]. These techniques preserve the basic structure of chitin while enhancing its crystallization index. Chitin nanofibers from *Mucor indicus* are also extracted from mechanical treatments and optimized culture conditions [78].

Fungal enzymes (e.g., from *Trichoderma harzianum*) were used to reduce chitosan to chitosan nanoparticles. These particles are spherical in shape and soluble at any pH. This process has also been implemented to produce fungal nanochitosans with appealing mechanical features for biomedical uses [78,79,80,81]. It shows potential for more eco-friendly production, although scaling up is generally tricky [80].

Nanochitosan synthesis practices from bacteria emphasize the development of eco-friendly approaches that allow the production of nanoparticles with specific features and stability. Compared with chitosan obtained from other commercial sources, bacterial-origin chitosan is highly biocompatible and biodegradable, and it has been reported to be used in biomedical applications without toxicity [82,83].

Once solubilized in acidic environments and vacuum-dried, chitosan reduces into nanoparticles with high functionalization possibilities, providing a high surface-to-volume ratio, higher drug-loading capacity, and enhanced antibacterial effect [84]. Nanochitosan has a strong antimicrobial effect, which is further improved with natural compounds/metals. As a result, it is active against many Gram-positive and Gram-negative bacteria [82,85].

Bacterial chitosan has been suggested as a green and environmentally friendly source. It is free from harsh chemicals and minimizes environmental effects, adhering to green practices [84,85].

## 4. Characterization of Nanochitin and Nanochitosan

Such methods are critical to grasp their potential use across various fields. Figure 5 depicts an overview of the different techniques used to characterize nanochitin.

### 4.1. Morphology and Size Distribution

Transmission electron microscopy (TEM) and scanning electron microscopy (SEM) characterization of nanochitin and nanochitosan is essential in understanding their properties and structure. Such solid-state characterization techniques further give insights into the nanoparticles’ size, shape, and distribution, which further helps apply nanoparticles in various sectors [86,87].

For instance, SEM and TEM images demonstrated a nearly spherical form and uniform dimension distribution with a mean dimension of ~42 nm in examining a magnetic guanidinylated chitosan nanobiocomposite. Homogeneity is vital for its application in catalysis and drug delivery [88]. The surface modifications of nanochitosan were also evaluated using SEM and TEM analyses. As depicted in the SEM and TEM images, the modification led to a rise in the amino functional groups on the surface of chitosan. Such functionalization is important for improving the reactivity and interaction of the material with other substances [88].

Dynamic light scattering (DLS) measures hydrodynamic size and polydispersity in solution [86,87] and has been applied to measure nanochitosan particle size. Nanochitosan extracted from waste shrimp shells and commercial chitosan was characterized, showing that nanochitosan generated particles with diameters lower than 223 nm and an average diameter of less than 25 nm. This indicates that the production method efficiently produces nanosized nanoparticles for various applications [87].

### 4.2. Crystallinity and Degree of Acetylation

X-ray diffraction (XRD) could be used for crystalline phase identification and crystallinity index calculation. XRD confirms the example of compounds encapsulated inside of the nanocomplexes, while FTIR aids in the identification of the interactions among molecules, such as hydrogen bonding and hydrophobic interactions. X-ray diffraction analyses demonstrate the high crystallinity of nanochitin and nanochitosan, highlighting their suitability for pharmaceutical applications in which stability and controlled degradation rates are critical [86].

Functional groups and acetylation degree determination are carried out by Fourier transform infrared spectroscopy (FTIR) and nuclear magnetic resonance (NMR) [89,90,91]. In the work mentioned above, the authors showed that 1H NMR is the most sensitive and accurate technique for determining DA in chitin and chitosan while providing precise data on the chemical structure of both biopolymers. 13C NMR is less sensitive than 1H NMR but is still analytically helpful and provides valuable information over the complete range of DA. FTIR spectroscopy effectively identifies characteristic functional groups within nanochitin and nanochitosan and confirms the success of chemical modifications critical for enhancing their pharmaceutical performance. NMR spectroscopy, including 1H and 13C NMR, provides precise and quantitative assessments of the degree of deacetylation. This measurement is crucial for determining biological activity and potential pharmaceutical applications of nanochitin and nanochitosan [5,6,7,8,9,10,11,12,13,14,15,16,17,18,19,20,21,22,23,24,25,26,27,28,29,30,31,32,33,34,35,36,37,38,39,40,41,42,43,44,45,46,47,48,49,50,51,54,55,56,57,58,59,60,61,62,63,64,65,66,67,68,69,70,71,72,73,74,75,76,77,78,79,80,81,82,83,84,85,86,87,88,89,90,91,92].

### 4.3. Thermal Stability

Thermogravimetric analysis (TGA) evaluates materials’ thermal degradation profiles [2]. TGA provides insight into chitosan’s thermal degradation and water adsorption capacity when used with techniques, such as FTIR and GCMS. This technique recognizes decomposition products, identifying physically and chemically adsorbed water molecules [93].

Among other methods, the TGA characterization of nanochitosan in biocomposites between carboxymethylcellulose and TiO_2_ helps understand the composite materials’ photocatalytic properties and thermal behavior. These characterization techniques collectively highlight the significant potential of nanochitin and nanochitosan for advanced biomedical and pharmaceutical formulations. Nonetheless, achieving consistency and reproducibility during synthesis remains a critical challenge for broader practical applications [94].

### 4.4. Molecular Weight and Viscosity

The properties of chitosan, including solubility, charge density, and encapsulation efficiency, primarily depend on its molecular weight. Higher molecular weight chitosan produces large, more positively charged amphiphilic nanocomplexes that can promote the encapsulation and stability of quercetagetin and other compounds [86,95].

Chitosan molecular weight is typically determined using asymmetrical flow field fractionation, light scattering detection (AF4-LS), size-exclusion chromatography, gel permeation chromatography (SEC or GPC), and viscometry. Calculating molar masses and the distribution of molecular weights is thus very important for understanding chitosan samples [95,96], and these methods can determine them.

Light scattering detection (AF4-LS) asymmetric flow field-flow fractionation (AF4) coupled with light scattering techniques is a powerful method for analyzing these nanomaterials, offering detailed insights into their size, molecular weight distribution, and aggregation behavior. It is beneficial because it can discriminate between chitosan molecules and aggregates in solution, enabling precise molecular weight distribution analysis [97,98].

Gel permeation chromatography (GPC) or size exclusion chromatography (SEC) [53] is a valuable tool for determining chitosan molecular weight distribution when developed with multi-angle laser light scattering (MALLS) and differential refractive index (RI) detectors. These polymers can be obtained free from molecular aggregates of the typical size found in chitosan solutions, making them suitable for molecular weight determination [97]. SEC-MALLS accurately characterizes chitosan at low injection concentrations and eluent flow rates, enhancing the ability to analyze molar mass distribution [99].

The SEC-MALLS method works for chitosan over a wide molar mass range (33 ≤ *M**u* ≤ 427 kg/mol). This means carefully controlling experimental conditions, such as low injection concentrations (0.1–0.2 mg/mL) to avoid column overloading, and the eluent flow rate should not exceed 0.5 mL/min for high molar mass samples [99].

Although not as precise, the method allows for the rapid determination of the change in molecular weight through viscometry [5]. The intrinsic viscosity of chitosan solutions is a characteristic of polymer molecular weight and solution conformation. The inherent viscosity, a critical parameter for characterizing polysaccharides, can be measured using capillary viscometry and models, such as the Huggins equation [100,101,102].

Temperature and molecular weight significantly impact the viscosity of chitosan solutions. High-molecular-weight chitosan has substantial flexibility and high intrinsic viscosities, decreasing linearly with increasing temperature. This behavior is consistent with temperature-induced conformational transitions outside the range studied [103].

## 5. Pharmaceutical Properties of Nanochitin and Nanochitosan

Due to their unique properties, nanochitin and nanochitosan have garnered significant interest, rendering them suitable for many applications, particularly in the pharmaceutical and biomedical domains.

### 5.1. Biocompatibility and Biodegradability

Due to their superb biocompatibility and biodegradability, nanochitin and nanochitosan are promising candidates for biomedical applications, including drug delivery and tissue engineering. Additionally, their natural origin and non-toxic nature make them a great candidate for use in a medical environment [7,8,58].

Since nanochitin is derived from nature and is biodegradable, it is considered the safest GRAS [42,104]. Its degradation products (primarily N-acetylglucosamine monomers) are generally non-toxic and capable of entering biological pathways [11].

### 5.2. Mucoadhesive and Bioadhesive Characteristics

Also, the amino and hydroxyl groups in nanochitin can form hydrogen bonds with the mucosal surfaces [105]. This property is vital to enhance the residence time of drugs in mucosal drug delivery systems [106]. The underlying mechanisms involved in the mucoadhesion of nanochitosan are illustrated in Figure 6, including ionic interactions, hydrogen bonding, and key chemical modifications that enhance adhesive strength.

Chitosan nanoparticles are characterized by strong mucoadhesive properties, making them helpful in developing drug-delivery systems directed to mucosal tissues. Chitosan’s mucoadhesive activity improves by increasing the water solubility and improving stability by chemical modification, such as the introduction of adhesion sites, such as trimethyl, carboxymethyl, and thiolated groups, at neutral and alkaline pH [107,108].

The selection of crosslinking agents influences the mucoadhesive properties of chitosan nanoparticles. For example, nanoparticles crosslinked with phytic acid (PA) and sodium hexametaphosphate (SHMP) show a significantly greater mucoadhesion potential compared with Agents crosslinked with TPP [109,110]. A critical factor for obtaining stable nanoparticles with strong mucoadhesive interactions is the ratio of chitosan to TPP. A 4:1 ratio provides stability and mucoadhesive strength, the minimum ratio for nanoemulsion formation [111].

When co-polymerized with other biomaterials, such as gum arabic and polyethylene oxide, chitosan-based nanoparticles increase adhesion to biomaterials, enhance drug absorption, and prolong drug retention at the site of action. This primarily benefits oral and sublingual drug delivery systems [112,113].

Chitosan derivatives, like glycol chitosan conjugates, have been explored to create mucoadhesive systems that may be used for intravesical drug delivery. These systems offer prolonged drug release and increased retention in the bladder [114]. The challenge is optimizing the nanoparticle’s size and surface properties for an adequate mucoadhesion and drug release balance. Optimizing these factors can produce more potent drug delivery systems [115]. Compared to other nano-polymers, such as poly (lactic-co-glycolic acid) (PLGA), alginate, and polyethylene glycol (PEG), nanochitosan exhibits superior mucoadhesive properties. Due to their neutral surface, PLGA nanoparticles are widely recognized for controlled drug release but possess lower intrinsic mucoadhesion. Alginate, negatively charged, also offers moderate mucoadhesion primarily via ionic interactions with mucin glycoproteins. PEG is commonly used for enhancing bioavailability through mucosal membranes; however, its mucoadhesion is limited by its hydrophilic and neutral characteristics. In contrast, the cationic nature of nanochitosan significantly enhances electrostatic interactions with negatively charged mucosal surfaces, providing higher mucoadhesiveness. This attribute makes nanochitosan particularly effective for drug delivery systems targeting buccal, nasal, and ocular mucosal membranes [107].

### 5.3. Antimicrobial, Antibacterial, and Immunomodulatory Effects

Nanochitin and nanochitosan also have strong antibacterial and antimicrobial properties. That is why they are beneficial in fields, like wound dressings and other medical gadgets where infection avoidance is needed [58,116]. Nanochitin has intense antimicrobial activity against Gram+ and Gram− bacteria [6,7]. It has also been studied in the context of immunomodulators, which could potentially be utilized in vaccine adjuvants.

Nanochitosan exerts its antimicrobial actions mainly by destroying microbial cell membranes. Because of its cationic property, it can interact with bacterial cell membranes that are primarily negatively charged and increase permeability, leading to leakage of intracellular components and cell death [82,117]. It has potent activity against Gram-positive and Gram-negative bacteria and fungi [83,117].

Factors such as particle size, molecular weight, and degree of deacetylation affect the antimicrobial effectiveness of nanochitosan. Particle sizes and molecular weights lower than this corresponding value have been shown to contribute to their improved antimicrobial activity [8,118]. Second, the antibacterial activities of chitosan are ascribed to the first amine groups [116,117].

The antimicrobial properties of nanochitosan against some pathogens, such as *Escherichia coli*, *Staphylococcus aureus*, and *Streptococcus pneumoniae*, have been well established [118,119]. Moreover, it has been effective against biofilms, frequently resistant to selective pharmacological drugs [117,120]. The ability to act synergistically with other antimicrobials, such as metallic nanoparticles or biosurfactants, improves bioactivity [83,120].

The mechanism of antifungal action by nanochitosan is shown in Figure 7. The antifungal activity of nanochitosan is primarily driven by its electrostatic interaction with the fungal cell membrane. Nanochitosan particles possess a positive surface charge (+) due to the protonated amino groups on their structure, particularly when the degree of deacetylation (DD) exceeds 80%. When nanochitosan is introduced into a biological environment, its positively charged particles are naturally attracted to the negatively charged fungal cell membrane. This membrane typically carries negative charges due to the presence of phospholipids and proteins. This strong electrostatic interaction leads to disruption of the membrane structure, compromising its integrity. This may involve pore formation, thinning of the bilayer, or complete rupture in localized regions. As the membrane integrity collapses, intracellular components, such as ions, proteins, and metabolites, begin to leak out of the fungal cell. This leakage causes a critical imbalance in homeostasis. The uncontrolled loss of essential cellular contents and the loss of membrane potential ultimately lead to irreversible cellular damage and fungal cell death. This mechanism is particularly effective against *Candida albicans* and *Aspergillus species*, and it explains the high antimicrobial efficacy of nanochitosan, especially when optimized to low molecular weight (<100 kDa) and high DD (>80%) [1,11,107].

Despite its potential, the formulation and delivery of nanochitosan for targeted applications still require optimization. Further studies are needed to understand better the interactions between nanochitosan and a wide variety of microbial species and to provide potential solutions for what might be resistance mechanisms [121,122].

## 6. Nanochitin and Nanochitosan and Their Pharmaceutical Applications

In Table 5, eight types of nanochitosan-derived nanomaterials are listed with their names, descriptions, applications, and references.

A graphical summary of the pharmaceutical applications of nanochitin and nanochitosan is presented in Figure 8, highlighting their roles in drug delivery, tissue regeneration, immunotherapy, antimicrobial systems, and theranostic platforms.

### 6.1. Nanochitin and Nanochitosan-Based Drug Delivery Systems

#### 6.1.1. Oral and Buccal Delivery

As already described, nanoparticles derived from chitosan have been a promising frontier in drug delivery applications owing to their biocompatibility, biodegradability, and ability to increase the bioavailability of drugs. Nanoparticles are especially relevant as oral and buccal drug delivery systems as they can help overcome several challenges associated with these delivery forms, including poor solubility and low bioavailability. Chitosan-based nanocarriers have effectively overcome the gastrointestinal barriers to administered molecules. In this way, they improve drug absorption and increase the oral bioavailability of a wide range of biotherapeutic products [150,151]. Another application of chitosan is buccal delivery systems to overcome the low bioavailability of some drugs, like curcumin, by avoiding first-pass metabolism and improving solubility to be more efficient. High-payload buccal films based on chitosan nanoplexes showed promising drug release profiles and adequate storage stability for buccal applications [152].

The mucoadhesive properties of nanochitin also improve its absorption across oral and buccal mucosa [105]. Such polymers can also protect active pharmaceutical ingredients from enzymatic degradation and increase their residence time [106]. Moreover, the stability of solid lipid nanoparticles coated with chitosan in acidic conditions enhances their applicability for oral drug delivery [153].

#### 6.1.2. Ocular Delivery

Chitosan is a non-toxic, biocompatible, and biodegradable cationic polysaccharide. It has mucoadhesive properties, which increase the adhesion of the ocular surface and consequently improve the residence time and bioavailability of the drugs. Nanochitin can be shaped into hydrogels or contact lenses, providing opportunities for the controlled release of ophthalmic drugs [154,155,156]. Mucosal vitamin foam works well in contact with the skin with low irritation and better patient comfort [136].

Current ocular delivery systems based on nanochitosan and nanochitin are proposed as innovative approaches to increase drug delivery to the eye, e.g., for age-related macular degeneration (AMD). These systems are utilized to enhance the potential of chitosan and chitin to maximize ocular therapeutics and minimize the ocular delivery systems’ approachability [157].

Nanotechnology-based platforms are emerging as potential solutions in ophthalmic therapeutics. They can improve treatment efficacy by modeling the unique barriers to treating pediatric eyes with retinoblastoma [158].

#### 6.1.3. Transdermal Delivery

Sustained transdermal drug delivery systems based on nanochitosan and nanochitin have attracted considerable interest due to their prospects in transdermal drug delivery. Utilizing the natural advantages of chitosan and chitin, such as biocompatibility and biodegradability, these systems enhance the penetration of drugs and their availability in the body. In addition, nanochitin is introduced into patches or microneedles to enhance the stratum corneum permeation of drugs by controlled release and stability [159,160].

Chitosan nanoparticles have demonstrated a high drug release rate, with formulations up to 60% release over several hundred hours, including 380 h [161].

New transdermal patch development, including the utilization of nanoneedles, has demonstrated increased permeant penetration. Some examples of structural geometry-based patches include pyramidal geometry nanoneedles with high overall drug permeation rates, such as 88.1% for levofloxacin [161].

#### 6.1.4. Targeted and Stimuli-Responsive Delivery

Specific target ligands (i.e., antibodies and peptides) can conjugate onto the surface of nanochitin particles to redirect their behavior and select their destination in organ tissues or cells. There is also active research on stimuli-responsive systems that disintegrate or release cargo in response to acidic or temperature conditions. Schools of thought have focused on developing systems responsive to multiple stimuli that signal changes in the circumambient environment. Chitosan-based microcapsules, for instance, have also been designed to be responsive to a magnetic field and reductive environment for targeted release and delivery of hydrophobic drugs. Folate-receptor-mediated targeting in these systems enhances targeting efficiency toward cancerous cells, indicating their potential biomedical applicability in the future [162]. In a similar approach, pH and glutathione (GSH)-responsive mesoporous silica nanoparticles coated with chitosan-based thin film have been developed to control drug release [163].

Adding a targeting moiety, such as folic acid, enhances the carrier’s selectivity to target cancer cells. This happens via folate receptor-aided endocytosis, which means higher cellular internalization and antitumor efficacy. These targeting strategies are critical in improving the specificity and effectiveness of chemotherapeutic agents [162,163]. Additionally, chitosan/hyaluronic acid nanoparticles were developed with responsiveness to oxidative stress and pH, endowing them with versatile, controlled-release properties [164].

Much has been investigated for their effectiveness in preclinical models through advanced delivery systems. Chitosan/hyaluronic acid nanoparticles, which are loaded with therapeutic agents, such as quercetin and curcumin, have been shown to kill glioblastoma cells. In addition, these nanoparticles retain the biological activity of proteins, such as nerve growth factors, promoting nerve outgrowth in vitro [164].

#### 6.1.5. Additional Formulations and Dosage Forms

In addition to the well-documented nanochitosan applications in oral, buccal, ocular, and transdermal delivery systems, other pharmaceutical dosage forms involving nanochitin and nanochitosan have been increasingly explored. These include mucoadhesive films and film-coated tablets that enhance drug retention in buccal or intestinal mucosa and offer improved stability for poorly soluble drugs [152]. Matrix tablets formulated with nanochitosan derivatives enable sustained drug release, particularly in gastrointestinal environments [150]. In aerosol and nasal spray formulations, nanochitosan has demonstrated promising results as a carrier for mucosal vaccine delivery due to its mucoadhesive and immunomodulatory properties [165,166].

Furthermore, suppositories and rectal delivery systems utilizing nanochitosan-based gels provide targeted drug release with enhanced bioadhesion, while thermoresponsive in situ forming gels and biogels have shown potential in ophthalmic and injectable applications [154,155,156]. Nanochitin hydrogels and nanocomposite gels have also been investigated for injectable wound-healing applications and cancer therapy due to their porous, responsive structure and excellent tissue compatibility [6,128]. These alternative dosage forms expand the pharmaceutical relevance of nanochitin and nanochitosan by enabling stimuli-responsive, site-specific, and patient-friendly delivery platforms, with growing interest in their scalability and regulatory readiness.

### 6.2. Tissue Regeneration and Wound Healing

Chitosan and chitin nanoformulation have shown great potential for wound healing. The nanometric characteristics of chitin enhance its natural hemostatic and antibacterial properties [149]. They also induce platelet activation and the upregulation of vascular endothelial growth factor, a crucial modulator of angiogenic tissue generation [167,168]. Chitin-based dressings act on nano-sized particles that promote faster wound closure, reduce infection risk, and support tissue regeneration. Other polymers, i.e., collagen and alginate, blended with nanochitin, have also been used to obtain higher mechanical strength and bioactivity [169].

### 6.3. Vaccine Adjuvants and Immunotherapy

The nature and immunomodulatory properties of nanochitin have evoked great interest in its use as a vaccine adjuvant [128]. Chitosan-based nanoparticles can increase the permeability of epithelial tight junctions, improving the uptake of particles and stimulating innate immune responses. This property is advantageous in mucosal vaccination as chitosan improves systemic and local immune responses by inducing antibodies and cytokines [165,166].

Chitosan-based nanoadjuvants have been shown to increase the immunogenicity of protein antigens via the nasal route. It is a non-invasive strategy for vaccine delivery as it can induce both humoral and cellular immune responses [165,166]. In mice, alginate-coated chitosan nanoparticles markedly boost the immunogenicity of the hepatitis A vaccine above a classical adjuvant, alum. This approach increases seroconversion rates and antibody levels, giving it a powerful potential for cost-effective vaccine production [170].

Chitin and its chitosan derivatives (e.g., glycated chitosan) have shown potential as adjuvants in cancer vaccines. These compounds can also modulate immune responses to diseases, making them applicable to enhance cancer immunotherapeutic outcomes [171]. Nanoadjuvant structure properties, such as size, surface charge, and morphology, play an essential role in establishing the effectiveness of cancer therapy. These parameters modulate the immune activity and safety of the adjuvants and highlight the need to optimize these factors for clinical applications [172].

### 6.4. Antimicrobial Formulations and Preservatives

Moreover, nanochitin exhibits superior antimicrobial activity to bulk chitin since it possesses an enormous surface area and can easily interact with the microbial cell walls. It is applied in pharmaceutical preservative systems, topicals, antimicrobial gels, and medical device coatings [42].

The combination of nanochitosan and nanochitin with other materials enhances their antimicrobial properties. For instance, chitosan nanoparticles have shown activity against bacterial infections through cell membrane damage, which promotes cell death [82,173]. Antimicrobially, these nanoparticles are efficient against Gram-positive and Gram-negative bacteria, and their effectiveness could be elevated by adding metallic nanoparticles or biosurfactants, such as rhamnolipid [83,120].

### 6.5. Tissue Engineering Scaffolds

Scaffolds based on nanochitosan and nanochitin are a target of interest in tissue engineering for regenerative medicine applications due to their potential to enhance the physical and biological properties of the scaffolds. It has been demonstrated that these biopolymers based on chitin and chitosan are especially interesting in bone tissue engineering and other fields because they are biocompatible, biodegradable, and can promote cell proliferation and differentiation. Nanochitin scaffolds provide a suitable microenvironment to promote cell adhesion, proliferation, and differentiation in regenerative medicine. By adjusting porosity, mechanical properties, and surface chemistry, these scaffolds can be tailored for specific tissues [54,174,175].

However, they are still limited in chitosan and chitin-based scaffolds as they need exact control over architecture and uniform performance in various applications. Optimal scaffold fabrication, including 3D printing and lyophilization methods, should be applied further to optimize these materials’ structural and functional properties [176]. Furthermore, when assessing the combined administration of nanochitosan or nanochitin with other biomaterials, their synergistic effects can lead to more effective scaffolds in tissue engineering applications [177].

Table 6 below shows a comprehensive detail on examples of commercial or near-commercial products that include nanochitosan (or chitosan in submicron scale) in their formulation.

## 7. Toxicological and Regulatory Considerations

Due to their unique characteristics, Nanochitin and nanochitosan have been widely investigated and are increasingly used in various fields, such as medicine, food packaging, and water treatment. Understanding their toxicological and regulatory implications is critical for ensuring safety and efficacy as their applications expand.

### 7.1. Toxicity Profile

Nanochitin is considered safe, but concerns are raised when that material enters the bloodstream or concentrates on specific organs [11]. Particle size, surface charge, and chemical modifications can affect biodistribution and cytotoxicity. More extensive studies are needed in vivo to prove long-term safety. For instance, chitosan-coated silver nanoparticles have eminent dose-dependent adverse effects on rats (50 mg/kg) where very striking toxicities occur [193].

Many studies have focused on nanochitosan for its cytotoxicity, particularly in the biomedical field. Results have shown that chitosan nanoparticles generally have low cytotoxicity independent of their composition and testing conditions. While this low cytotoxicity makes them a promising candidate for medical applications, each new derivative must be thoroughly characterized and evaluated for cytotoxicity before market implementation [83,194,195].

In the context of pharmaceutical and biomedical applications, the potential contamination of nanochitin and nanochitosan materials with nitrosamines and heavy metals represents a serious safety concern. Nitrosamines, known for their carcinogenicity, may arise from the interaction of amine groups with nitrosating agents during manufacturing or storage. Similarly, heavy metals, such as lead, cadmium, arsenic, and mercury, can be introduced through raw materials, extraction reagents, or equipment [1,11].

To ensure safety, stringent analytical methods are required to detect and quantify such contaminants. Techniques, such as inductively coupled plasma mass spectrometry (ICP-MS) and liquid chromatography–mass spectrometry (LC-MS/MS), are widely recommended for their sensitivity and reliability. Regulatory agencies, including the U.S. Food and Drug Administration (FDA) and the European Medicines Agency (EMA), mandate strict limits and regular monitoring of these substances in pharmaceutical products. Incorporating validated contaminant screening protocols during raw material sourcing, processing, and final formulation is essential to minimize toxicological risks and ensure compliance with current regulatory standards [1,11].

### 7.2. Regulatory Framework

Chitosan nanoparticles are characterized by low cytotoxicity, which makes them suitable for biomedical research and applications. Nevertheless, novel chitosan derivatives and compositions must be well-characterized, especially for cytotoxicity, before being available on the market [195].

Thus, pharmaceutical nano-based materials are regularly reformulated internationally [196]. Manufacturers must prove that nanochitin meets rigid standards concerning purity, consistency, and biocompatibility. Regulatory authorities, like the European Medicines Agency (EMA) and the U.S. Food and Drug Administration (FDA), are increasingly requiring standardized characterization procedures and comprehensive risk assessments [104,197].

### 7.3. Environmental and Ethical Aspects

The sustainable sourcing of chitin, mainly through crustacean waste, aligns with the principles of the circular economy [198]. Nevertheless, scale acid hydrolysis or chemical treatments could cause environmental issues. Using conventional methods raises the ecological footprint, but more environmentally friendly or enzymatic methods minimize this requirement [18].

The production of nanochitin and nanochitosan and their application have positive environmental and ethical benefits. The development of these materials is consistent with the need for green manufacturing and ethical assets usage to make them a developing material in various industries [7,41,60,92,199,200].

### 7.4. Physical and Chemical Stability

A crucial consideration in the design of nanochitin- and nanochitosan-based pharmaceutical formulations is their physical and chemical stability under diverse environmental and processing conditions. These nanoscale biopolymers exhibit enhanced structural stability due to their high surface area and strong intermolecular interactions. However, they remain sensitive to external stressors, such as pH variation, temperature shifts, humidity, and ionic strength [11,93].

Stability against heat, gamma irradiation, and autoclaving is especially relevant for sterilization and storage of injectable or implantable systems. Several studies have demonstrated that nanochitin cryogels and nanochitosan hydrogels retain their morphology and performance post-sterilization, particularly when chemically stabilized using crosslinkers, such as genipin, glutaraldehyde, or tripolyphosphate (TPP) [6,11,128]. Thermal stability evaluations using thermogravimetric analysis (TGA) and differential scanning calorimetry (DSC) confirm that nanochitin cryogels exhibit predictable degradation patterns, maintaining their structure under temperatures compatible with biomedical applications [94,149]. Moreover, nanoformulations, such as nanoparticles, nanogels, and emulsions, are subject to stability assessments using zeta potential and dynamic light scattering (DLS) to monitor aggregation tendencies, shifts in particle size, and surface charge modifications during storage or under mechanical stress. The inclusion of surface coatings, polyelectrolyte complexes, or lyophilization protocols improves shelf-life by minimizing aggregation and protecting functional groups [87,105,107].

Importantly, stability has been assessed in various physiological environments, including acidic (pH~1.5–3) gastric fluid, neutral mucosal environments (pH~6.5–7.4), and biological fluids with elevated ionic strength. Nanochitosan’s pH-dependent charge behavior contributes to its mucoadhesive retention at low pH, while nanochitin demonstrates resistance to swelling and degradation in neutral to slightly alkaline conditions. These differences underscore the importance of matching formulation types to the intended route of administration, particularly in oral, nasal, ocular, or injectable systems.

Figure 9 shows the physical and chemical stability of nanochitin and nanochitosan under environmental and processing conditions.

## 8. Looking Ahead: Issues and Perspectives

Nanoscale chitin and chitosan have great potential for biomedical devices, catalysis, and therapeutics due to their biocompatibility, antibacterial activity, and immunogenicity. Nonetheless, nanochitin and nanochitosan production is a great challenge, an opportunity for sustainable development, and advanced material applications in the future [8].

### 8.1. Scalability and Cost-Efficiency

Even when lab-scale results are promising, such cost-effective, up-scale processes to produce nanochitin are still challenging. Research on continuous-flow reactors, enzyme-based extraction, and the recycling of reaction media may optimize production and reduce costs.

Nanochitosan must be produced on a larger scale. Affordability and simplicity are potentially scale-up technologies, with promising results at both bench and pilot scales, such as those obtained through thermal shock and ball milling. In addition to being low energy-consuming, microwave-assisted extraction leads to a rapid route for generating high-molecular-weight and low-acetylation chitosan. Acid-assisted colloid milling, another efficient commercial method that combines glycerol swelling and a colloid mill, enabled high yields and excellent dispersibility, features especially useful on an industrial scale [61,87,201].

Various approaches exist to improve cost efficiency. One is to use waste shrimp and crab shells as raw materials, reducing raw material costs and supporting sustainability. Reducing processing time and the energy-saving capabilities of microwave technology add to the cost-friendly production of chitosan. Large-scale production has also been assessed via exergy analyses to determine where process efficiency may be improved, mainly via precision in washing and drying [10,87,201,202].

### 8.2. Standardization and Quality Control

Direct comparisons among studies are difficult because of variations in extraction methodologies, degrees of acetylation, and molecular weight [5]. Characterization of nanochitin could also include crystallinity index, zeta potential, and safety testing according to standardized protocols, which will further ease their regulatory approval and commercial adoption [197,203]. The absence of standardization and quality control in their manufacturing procedures challenges their safe and effective utilization. In developing these materials for pharmaceutical applications, accurate reporting of reproducibility is critical [204].

There is a need for standardized preparation and characterization methods for chitosan and its derivatives. This absence of standards hinders the consistent implementation of the Safe-by-Design (SbD) framework, which is necessary to ensure the safety and efficacy of nanomedicines [87,204]. Quality control requires proper characterization and documentation of chitosan properties. This makes it challenging for nanochitosan products to be reproducible and reliable, which could affect their performance as a drug carrier in most drug delivery applications [204].

### 8.3. Multifunctional Systems

Designing multifunctional nanochitin platforms is enabled by the convergence of nanotechnology, materials science, and pharmaceutical chemistry.

Also, 3D printable nanochitin bioinks—research way. Nanochitin bioinks are highly stretchable, super-flexible, and lightweight, making them suitable for a broader range of applications in modular 3D printing, such as flexible electronic devices and tissue scaffolding. Their ultrathin nature enables accurate and controlled placement during 3D printing, which is essential for manufacturing detailed and complex architecture [149,205,206]. Moreover, modifying bioinks through functionalization is a decisive factor in 3D bioprinting for fabricating biophysical-functional tissues and organs to overcome current material limitations [207].

Innovative responsive wound dressings with drug delivery are a promising development in wound care that can enrich remedies regarding wound healing, such as chitin and nanofibers of chitosan. These dressings are designed to respond to environmental stimuli, with the potential for targeted and effective treatment of different types of wounds, even those infected with drug-resistant bacteria. Usually, pH-responsive materials are addressed in innovative wound dressings to control drug release. An example is the pH-sensitive curcumin release of nanochitosan-reinforced polycaprolactone membranes, which are promising for wound dressing with modifiable drug release properties [59,208]. Correspondingly, chitosan-polyethylene oxide/polycaprolactone nanofibrous mats exhibited a sequential release profile of loaded lidocaine hydrochloride and curcumin, which reacted under acidic conditions and acquired a rapid-release and long-term effect of analgesic and antibacterial properties, respectively [209].

Then, there are the delivery vehicles for gene-editing CRISPR/Cas9 systems. The key obstacle to successfully using CRISPR/Cas9 systems for gene editing in therapeutic settings is to devise efficient delivery vehicles. Chitosan derivatives, nanochitin, and nanochitosan have attracted attention as non-viral delivery carriers because of their biocompatible properties and ability to form stable nanocomplexes with nucleic acids. Chitosan as a biopolymer has many advantages in gene delivery, such as forming stable nanocomplexes with nucleic acids because of its positively charged amino groups. This characteristic allows the formation of strong electrostatic interactions with the anion components, which subsequently promotes their delivery into target cells. Nonetheless, the low solubility of chitosan at physiological pH and its weak buffering capacity are issues that chemical modifications can solve [210,211,212,213].

### 8.4. Personalized Medicine and Theranostics

Theranostics, or personalized medicine, is a new paradigm in public health that adds a unique aspect to each individual and the available treatments. Nanochitin would also be helpful in customized medicine strategies, integrating diagnostic and therapeutic functions. This surface conjugation with imaging agents or biosensors could enable real-time monitoring of drug efficacy or pathological changes at the target site. Personalized medicine and theranostics still represent a game-changer in healthcare by focusing on individualizing medical protocols depending on the characteristics specific to each patient [57,166,214].

Chitosan contains functional groups that allow for facile chemical modification, increasing the applicability of this polymer in multifunctional nanosystems for targeted therapy and diagnostics [57]. Later, integrating diagnostic and therapeutic functions into a single system called theranostic agents showed promise for personalized medicine with minimal side effects [215].

The schematic flowchart shows in Figure 10 illustrates the integral stages in the transformation of natural sources of chitin into nanochitin and nanochitosan-based pharmaceutical materials. The process begins with biogenic sources, such as fungi, crustaceans, and insect biomass, followed by extraction via methods, like acid hydrolysis, ionic liquids, or deep eutectic solvents. These methods break down the crystalline structure and deacetylate chitin to yield chitosan or nanoscale derivatives. The extracted nanostructures undergo chemical functionalization to tailor surface properties for targeted delivery or enhanced bioactivity. Subsequently, physicochemical characterization is performed to assess particle size, morphology, zeta potential, and drug encapsulation efficiency. These steps culminate in advanced pharmaceutical applications, including drug delivery systems, wound dressings, immunoadjuvants, and theranostic platforms. The figure underscores the interdisciplinary and translational nature of nanochitin and nanochitosan technologies bridging natural polymers and therapeutic innovation.

## 9. Conclusions

This comprehensive study highlights the remarkable potential of nanochitin and nanochitosan for various pharmaceutical and biomedical applications. Their inherent biocompatibility, biodegradability, and chemical versatility make them strong candidates for advanced drug delivery systems, wound dressings, and immunomodulatory platforms. Furthermore, their nanoscale size and high surface area enhance mechanical properties and mucoadhesive abilities, enabling more precise and sustained delivery of therapeutic agents across different biological barriers. For pharmaceutical applications, fungal- and crustacean-derived chitin are the most suitable due to their high purity, well-documented properties, and scalability. Fungal sources are particularly advantageous for medical and vegan products due to their biocompatibility and low contamination risk. Crustacean sources remain dominant due to availability and processing efficiency but require rigorous purification to meet regulatory standards.

Despite significant progress, challenges remain. Standardized protocols for extraction and characterization are crucial to ensure consistent quality, enhance regulatory approval, and facilitate large-scale production. Equally important is addressing potential environmental impacts by adopting greener processes and utilizing renewable raw materials. Additionally, long-term in vivo studies are necessary to confirm safety and efficacy, particularly regarding the influence of particle size and chemical modifications on toxicity.

The field benefits from multidisciplinary research that connects materials science, engineering, and clinical medicine. Innovations, such as responsive nanocomposites, personalized drug delivery platforms, and 3D-printable bioinks, exemplify the versatility of nanochitin-based systems. By establishing rigorous quality controls and refining sustainable extraction methods, nanochitin and nanochitosan will likely play a pivotal role in next-generation therapeutic strategies and regenerative medicine.

## Figures and Tables

**Figure 1 pharmaceutics-17-00576-f001:**
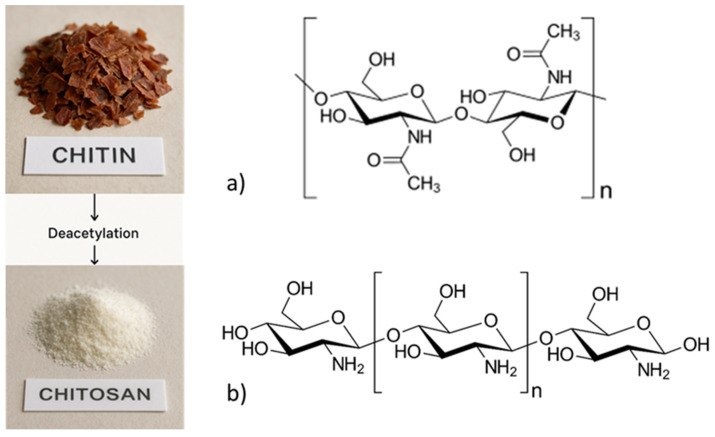
Chitin from crustacean shells (**a**) and chitosan (**b**) chemical structures.

**Figure 2 pharmaceutics-17-00576-f002:**
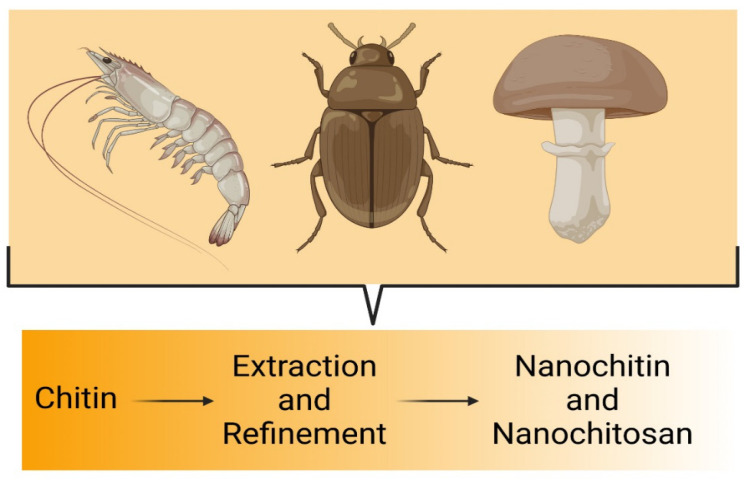
Natural products to obtain nanochitin and nanochitosan.

**Figure 3 pharmaceutics-17-00576-f003:**
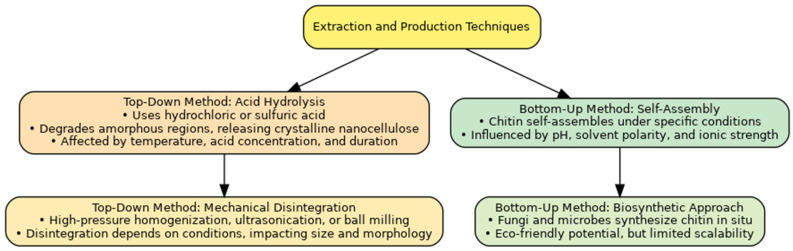
Nanochitin and their production techniques.

**Figure 4 pharmaceutics-17-00576-f004:**
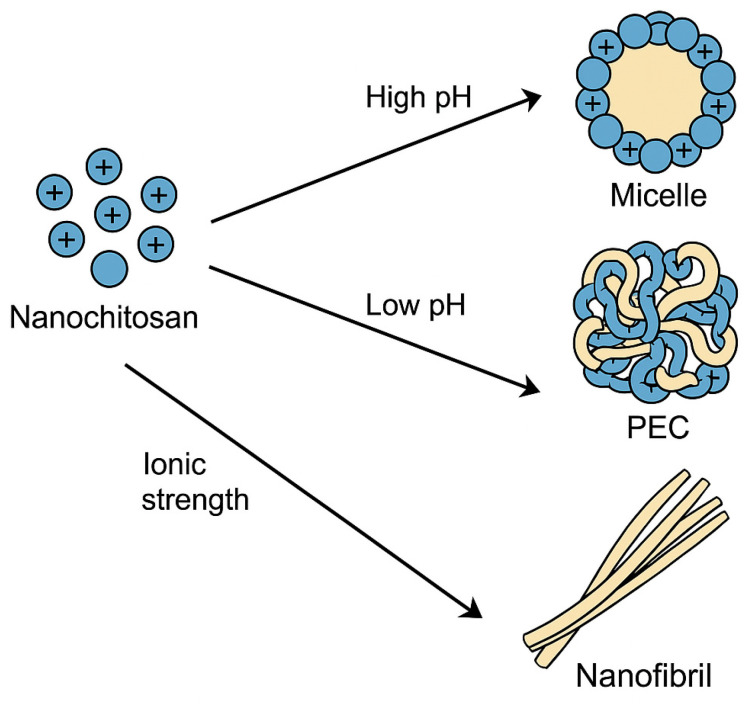
The schematic diagram presents micelle formation, corresponding to mechanisms that are critical for drug delivery and biomedical applications [71,73].

**Figure 5 pharmaceutics-17-00576-f005:**
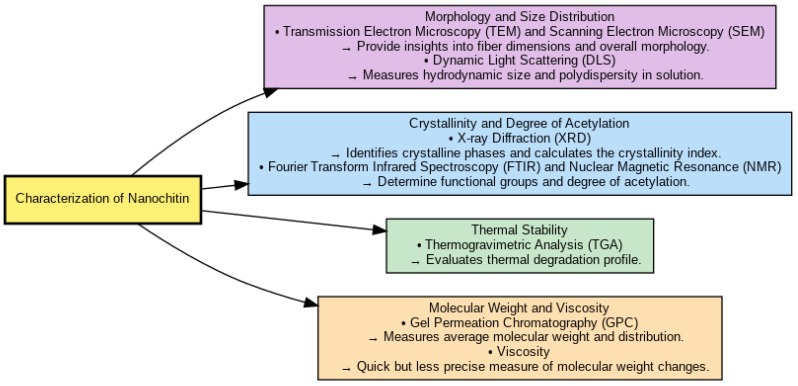
Nanochitin characterization techniques.

**Figure 6 pharmaceutics-17-00576-f006:**
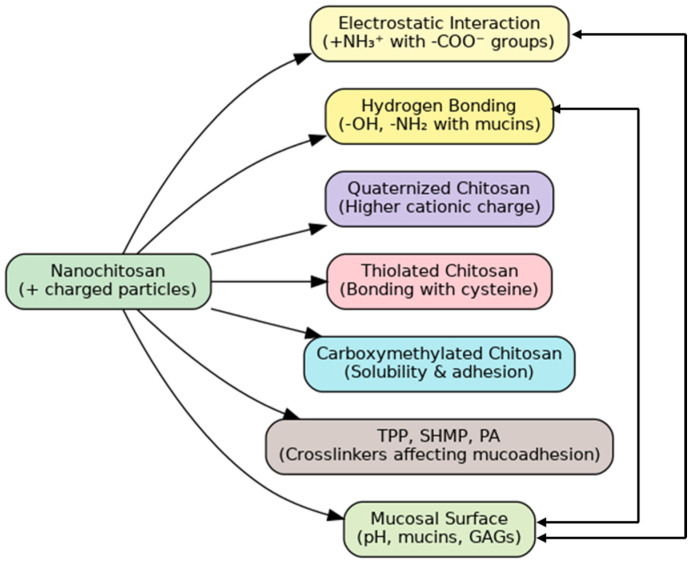
Mechanistic diagram of mucoadhesion by nanochitosan particles. Positively charged groups interact with mucosal surfaces via hydrogen bonding and electrostatic forces. Chemical modifications, such as quaternization, thiolation, and carboxymethylation, further enhance bioadhesion, supported by various crosslinking agents.

**Figure 7 pharmaceutics-17-00576-f007:**
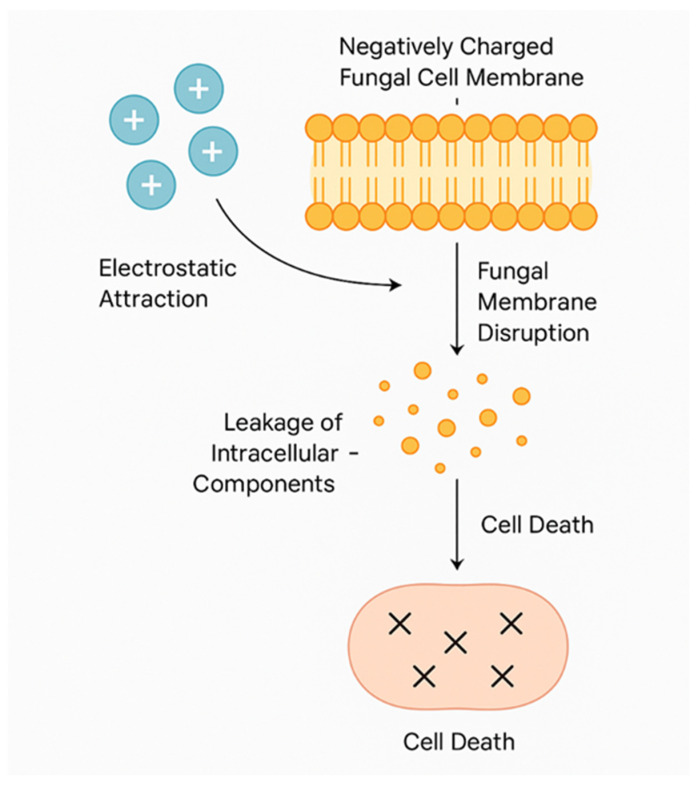
Mechanism of antifungal action by nanochitosan.

**Figure 8 pharmaceutics-17-00576-f008:**
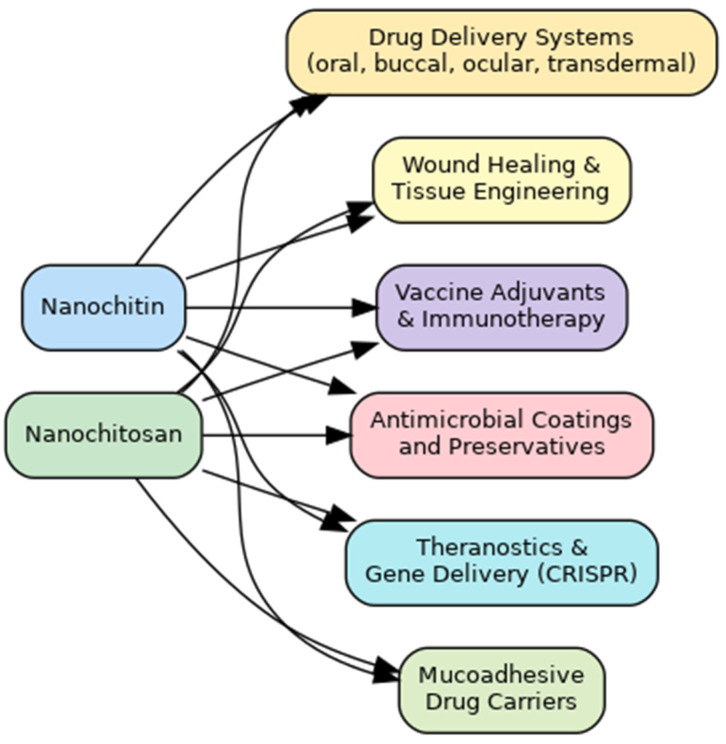
Schematic overview of major pharmaceutical and biomedical applications of nanochitin and nanochitosan. These include drug delivery systems (oral, buccal, ocular, transdermal), tissue engineering, immunoadjuvants, antimicrobial coatings, theranostics, and mucoadhesive platforms.

**Figure 9 pharmaceutics-17-00576-f009:**
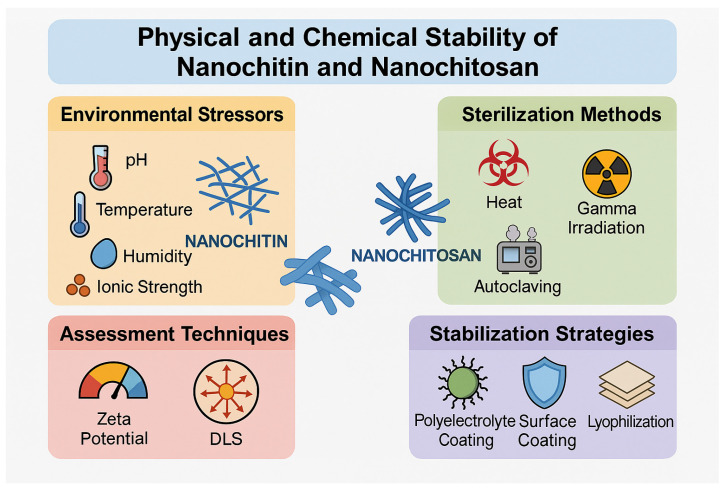
Physical and Chemical Stability of Nanochitin and Nanochitosan Under Environmental and Processing Conditions.

**Figure 10 pharmaceutics-17-00576-f010:**
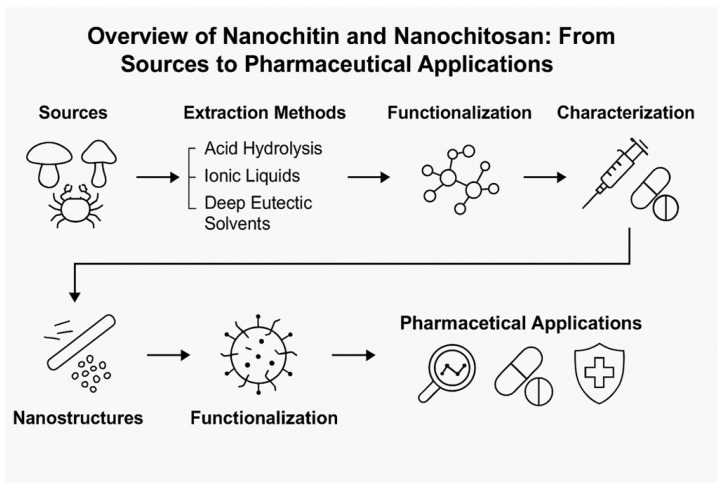
Nanochitin and nanochitosan: form sources to pharmaceutical applications.

**Table 1 pharmaceutics-17-00576-t001:** Species of fungi that produce chitin.

Species of Fungi	Information	Reference
*Aspergillus* species	Chitin synthesis in *Aspergillus* is well-documented, with up to eight synthase-encoding genes identified. This highlights its significant role in fungal growth and interaction with the environment.	[24]
*Candida albicans*	This species exhibits high chitin content, making it a notable producer among pathogenic fungi.	[25]
*Cryptococcus gattii* y *Aspergillus niger*	Both species are recognized for their high chitin production, like *Candida albicans*.	[25]
*Fusarium* species	*Fusarium* KYM3 is noted for its high chitin yield, making it a prominent producer among soil fungi.	[26]
*Penicillium* species	Although *Penicillium* KYM6 is noted for lower chitin production compared to *Fusarium*, it is still a significant source.	[26]
Zygomycetous fungi	*Mucor rouxii* and *Rhizopus oryzae* are extensively studied for chitin and chitosan production.	[27]
Basidiomycetes	Includes fungi, like *Agaricus bisporus*, which are known for their chitin content and are used in various applications.	[27]

**Table 2 pharmaceutics-17-00576-t002:** Chitin and chitosan solubility profile.

Solvent	Chitin	Chitosan	References
Dilute Acetic Acid	Insoluble	Soluble	[1,5]
Concentrated Acetic acid	Partially soluble	Soluble	[52]
Water	Insoluble	Insoluble	[1,6]
Hexafluoroisopropanol	Partially	Soluble	[11,28]
Ionic Liquids	Soluble	Soluble	[6,53]
Deep Eutectic Solvents	Soluble	Soluble	[28,33]
Lactic Acid	Partially	Soluble	[6,44]
Formic Acid	Partially	Soluble	[1,18]
Citric Acid	Partially	Soluble	[12,17]
Malic acid	Insoluble	Slightly soluble	[52]
Glycolic acid	Insoluble	Slightly soluble	[52]
EDTA and chelators	Facilitates swelling	Facilitates partial dissolution	[32,33]

**Table 3 pharmaceutics-17-00576-t003:** Chitosan derivates and their uses and properties [1,49,54].

Derivative	Modification	Properties	Applications
N-Carboxymethyl Chitosan (NCMC)	Introduction of carboxymethyl groups (-CH_2_-COOH) to the amino groups.	Improved water solubility, biocompatibility, and chelating ability.	Wound healing, controlled drug delivery.
Chitosan Sulfates	Sulfation of hydroxyl or amino groups to introduce sulfate groups (-OSO_3_H).	Enhanced anticoagulant and antiviral activity.	Anticoagulant materials, antiviral agents.
Quaternized Chitosan (QCS)	Introduction of quaternary ammonium groups (-N⁺(CH_3_)_3_) to the amino groups.	Increased water solubility, strong cationic nature, enhanced antimicrobial activity.	Antimicrobial coatings, gene delivery systems.
Thiolated Chitosan	Introduction of thiol groups (-SH) to the chitosan backbone.	Improved mucoadhesive properties and enhanced drug permeation.	Mucoadhesive drug delivery systems, wound healing materials.
Chitosan Oligosaccharides (COS)	Enzymatic or chemical hydrolysis to produce low molecular weight oligomers.	Enhanced solubility, bioavailability, and antioxidant activity.	Nutraceuticals, plant growth promoters, antifungal agents.
Grafted Chitosan	Grafting of synthetic polymers (e.g., polyethylene glycol) onto the chitosan backbone.	Tunable mechanical and thermal properties improved hydrophilicity or hydrophobicity.	Tissue engineering scaffolds, controlled drug delivery systems.

**Table 4 pharmaceutics-17-00576-t004:** Chitin and chitosan have enhanced properties in the nanoscale [12,48,55].

Property Enhanced	Description	Applications
Mechanical Properties	Due to increased surface area and intermolecular interactions, nanosizing improves tensile strength, elasticity, and durability.	Nanofibers and nanocomposites for tissue engineering, wound dressings, and biodegradable films.
Colloidal Stability	Nanoparticles exhibit better dispersion and stability in aqueous solutions, preventing aggregation.	Drug delivery systems, water treatment, and food packaging.
Biocompatibility and Biological Interactions	Nanoparticles enhance interactions with cells and tissues, improving biocompatibility and promoting bioactivity.	Tissue engineering, wound healing, and regenerative medicine.
Drug-Loading Capacity and Release Profiles	The high surface area of nanosized particles allows for higher drug-loading capacity and controlled release kinetics.	Cancer therapy, antimicrobial delivery, and sustained-release formulations.

**Table 5 pharmaceutics-17-00576-t005:** Categories of nanomaterials derived from nanochitin and nanochitosan.

Nanomaterial	Description	Applications	References
Ionic Gelation Nanoparticles	Formed by electrostatic interaction between acidic chitosan solution and polyanions (e.g., sodium tripolyphosphate, TPP). Produce stable nanosized particles.	Controlled drug release in oral or parenteral systems. Protection of proteins and peptides against enzymatic degradation.	[123,124,125,126,127]
Chitosan Nanogels	Three-dimensional polymeric networks at the nanoscale that retain significant amounts of water or physiological fluids. Can be synthesized via chemical or physical gelation (e.g., pH or ionic changes).	Stimuli-responsive release (pH, temperature, etc.). Potential for ocular, dermal, and transmucosal formulations owing to high biocompatibility. Nanogels loaded with doxorubicin have shown sustained drug release, which can be enhanced under specific conditions, like near-infrared laser irradiation and acidic pH.	[128,129,130]
Electrospun Nanofibers	Obtained through electrospinning of chitosan-based solutions (often blended with PVA, PLA, or other polymers). Produce membranes with high porosity and surface area. AgNPs are incorporated through in situ synthesis in the spinning solution, which ensures uniform dispersion within the nanofibers.	Wound dressings featuring antibacterial and hemostatic properties. Tissue engineering scaffolds promote cell adhesion and proliferation. They are particularly effective in wound dressings, where they can absorb exudates and inhibit microbial growth. Adding AgNPs enhances their antibacterial activity, making them suitable for treating infections caused by bacteria, such as *Pseudomonas aeruginosa* and methicillin-resistant *Staphylococcus aureus* (MRSA). Adding materials, like g-C_3_N_4_/TiO_2_, can enhance the photocatalytic properties of chitosan nanofibers, improving their efficiency in pollutant removal under visible light.	[131,132,133,134,135]
Nanocapsules and Nanoemulsions	Colloidal systems in which chitosan acts as an emulsifier or coating. These systems leverage the unique properties of chitosan, a natural biopolymer, to improve the encapsulation and delivery of various substances, including curcumin and peptides. Often oil-in-water (O/W) emulsions at the nanoscale.	Enhanced solubility for lipophilic drugs. Targeted delivery (via surface modifications) for improved therapeutic outcomes. Chitosan nanoemulsions and nanocapsules are used in the food and pharmaceutical industries to improve the solubility, stability, and bioavailability of hydrophobic compounds, like curcumin and eugenol.	[136,137,138,139]
Layer-by-Layer (LbL) Polyelectrolyte Nanoparticles	Built by alternately depositing cationic (chitosan) and anionic (e.g., alginate, carrageenan) layers. Yield multilayered nanoparticles or coatings with tunable properties.	Sequential or pulsatile drug release. Fine control over surface properties and responsiveness (e.g., pH, ionic strength). Chitosan/dextran sulfate/chitosan (CS/DEX/CS) nanoparticles have been developed for dual drug delivery. They demonstrate controlled release profiles and enhance cytotoxic effects against cancer cells. Similarly, chitosan nanoparticles have been used to create drug-release coatings on PCL nanofibers, showing potential for therapeutic protein delivery.	[140,141,142]
Chitosan–Metal Nanocomposites	Incorporate metallic nanoparticles (e.g., silver, gold, zinc oxide) dispersed in the chitosan matrix. Combine the bioactivity of chitosan with the antimicrobial or catalytic properties of metals. Typical metals used include silver, gold, platinum, and palladium. Silver nanoparticles are generally more significant than others, leading to different morphologies in the resulting films.	Enhanced antimicrobial performance for medical device coatings and wound dressings. Potential in biosensors or catalytic applications due to combined biocompatibility and metallic functionality.	[143,144,145]
“Nanogel-in-Microsphere” Hybrid Systems	Hybrid structures where chitosan nanoparticles are encapsulated within microparticles made of another polymer, or vice versa. Enable multiple modes of drug release.	Dual drug release (for hydrophilic and hydrophobic molecules). Applications in vaccines and gene therapy (protection and controlled release of DNA or RNA). Chitosan-based hybrid nanogels with covalent crosslinking show excellent stability and reversible pH response, combining multiple functions into a single nano-object for biomedical applications.	[146,147]
Drug–Nanochitosan Conjugates	Covalent or electrostatic conjugates between nanochitosan and drugs, proteins, enzymes, or antibodies. Often include ligands for tissue or cell-specific targeting.	Targeted therapy via site-specific delivery to cells or organs. High intracellular efficacy for delivering oligonucleotides, siRNA, etc.	[2,50,148]
Nanochitin Cryogels	Highly porous, sponge-like networks formed via freeze-drying of nanochitin hydrogels.	Wound healing, tissue regeneration, scaffolds for 3D cell culture.	[6,9]
Nanofibrillar Films	Thin films are composed of aligned nanochitin fibrils with high mechanical strength and water retention.	Antimicrobial wound dressings, drug release matrices, packaging for biomedical products.	[65,149]
β-Chitin Nanowhiskers	Rod-shaped nanostructures from squid pens with enhanced surface area and reactivity.	Controlled drug delivery, mucoadhesive systems, injectable depots.	[6,75]

**Table 6 pharmaceutics-17-00576-t006:** Examples of commercial products include chitosan.

Commercial Name	Formulation	Indication	Product Type	Observations	References
ChitoTech Hemostatic Dressing	Hemostatic product based on chitosan. Utilizes nano/micro-scale chitosan particles with high adsorption capacity.	Rapid control of bleeding in acute or traumatic wounds. Infection prevention.	Medical device (hemostatic dressing)	It is commercially available for emergencies and hospital use. The manufacturer highlights “nano/micro chitosan” to enhance adhesion and hemostasis. It is offered in various sizes for different wound types.	[178,179]
ChitoGauze^®^ (Tricol Biomedical/HemCon)	Gauze impregnated with submicron chitosan. Designed to adhere to bleeding sites and enhance coagulation.	Emergency hemorrhage control in trauma. Used in both military and civilian settings for external bleeding.	Medical device (hemostatic dressing)	FDA-cleared in the U.S. via 510 (k) as a hemostatic dressing. Although “nano” is not explicitly labeled, the chitosan is reportedly present in reduced particle sizes for increased contact area and faster action.	[180,181,182,183,184]
Chitoderm^®^ Gel	Gel containing “nanochitosan” or “oligochitosan” to improve penetration and antimicrobial effect.	Treatment of minor burns, diabetic foot ulcers, and pressure sores. Promotes wound healing.	Medical device/wound care product	Marketed in certain regions (primarily Asia/Eastern Europe) as a topical gel. It claims superior performance due to higher surface area from nanochitosan, yet there is limited public data on exact particle size.	[185]
Nasal Sprays with nanochitosan (various brands)	Aqueous solution with chitosan nanoparticles or microcapsules to improve mucosal adhesion in the nasal cavity.	Alleviates nasal congestion and allergic rhinitis or serves as a protective barrier. It may enhance the absorption of nasal drugs or supplements.	Medical device/nutraceutical (depending on jurisdiction)	The exact composition and concentration vary by manufacturer. Often registered as “medical devices” (CE) or supplements. Reported to have good tolerance and prolonged residence on the nasal mucosa.	[186,187,188]
“Nano-Chitosan Fat Binder” Supplements (various)	Capsules or powders claimed to contain nano-sized (or submicron) chitosan for enhanced fat-binding capacity.	Weight management and dietary cholesterol reduction. Marketed as adjuncts to weight-control regimens.	Nutritional supplement/nutraceutical	Available under different brand names, often without extensively validating proper nanoscale dimensions. Loosely regulated; classified as “dietary supplements” in some countries. Efficacy and bioavailability may vary widely across products.	[136]
mRNA/siRNA Delivery Formulations (early clinical)	- Vesicles or chitosan nanoparticles modified to encapsulate nucleic acids (RNA, siRNA, mRNA).	Gene therapy and next-generation vaccination. Protects genetic material from enzymatic degradation and improves cellular uptake.	Experimental formulations in preclinical/early clinical trials	Not yet widely commercialized. Smaller biotech companies are exploring pulmonary, nasal, or oral administration prototypes. Require rigorous regulatory approval to be recognized as “drugs.”	[128,189]
Ocular Implants/Gels (Prototypes)	Gels and microcapsules containing nanochitosan to prolong ocular drug release.	Treatment of ocular diseases (glaucoma, infections, etc.) with sustained drug release.	Preclinical/clinical research formulations	It is still in the testing phase; it is not broadly available on the market. It is claimed that nanochitosan ensures good corneal bioadhesion and improves drug bioavailability.	[106,190,191,192]

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
