# Peer review of "Nanochitin and Nanochitosan in Pharmaceutical Applications: Innovations, Applications, and Future Perspective"

_pharmaceutics, 2025, doi:10.3390/pharmaceutics17050576_

Round 1

Reviewer 1 Report

Comments and Suggestions for Authors

Chitosan is extensively employed in pharmaceutical fields, as it is bioadhesive, biodegradable and non-toxic polymer. The review is of substantial interest in the field of nanopharmaceutics, sufficiently original and logically reasonable. I recommend manuscript for publication after addressing the following.

Section 3.3.1.1 describes chitosan-based interpolyelectrolyte complexes. Nowadays the new type of IPECs  is delievered, where both components of the complex are biodegradable . It would be useful to consider such IPECs in more detail, including their applications, perspectives and disadvantages.

Author Response

Reviewer 1

Comment:

"Section 3.3.1.1 describes chitosan-based interpolyelectrolyte complexes. Nowadays, the new type of IPECs is being delivered, where both components of the complex are biodegradable. It would be useful to consider such IPECs in more detail, including their applications, perspectives, and disadvantages."

Response:

The following paragraph was added to Section 3.3.1.1:

Recent developments highlight these interpolyelectrolyte complexes (IPECs) consisting of fully biodegradable components, which significantly enhance biocompatibility and environmental sustainability. Applications of biodegradable IPECs include controlled drug release systems, targeted tissue engineering scaffolds, and innovative biomedical formulations. Notable advantages encompass reduced environmental impact, improved biocompatibility, and enhanced biodegradability, making them suitable for medical and pharmaceutical applications. Nevertheless, these biodegradable IPECs face certain challenges such as limited stability under specific physiological conditions, potential rapid degradation rates, and difficulties related to scalability and reproducibility in industrial production processes. Addressing these limitations is crucial for broader clinical and commercial applications.

Reviewer 2 Report

Comments and Suggestions for Authors
  1. Title, please consider using the word as nanochitosan or Nanochitosan? and please reconsider the suitability of using "in Pharmacy" in the title since it seems too generalized meaning.
  2. Please modify Abstract into single paragraph and it is found that there is no description on nanochitosan in Abstract?. For last paragraph, that should be chitin or nanochitin and nanochitosan?
  3. Line 110, 144, Chitosan and Chitin should be chitosan and chitin; Line 111, 487-488 please change to italic style of scientific name of microbe.
  4. Please include review chitin from other source including squid and snail.  
  5. Please include solubility in different solvents of chitin and chitosan in 2.2. 
  6. Line 220, Microwave Irradiation should be microwave irradiation.
  7. Line 259, what is the full name of TCP.
  8. For topic 3, please include the technique from your review from the patent.
  9. For 3.3.1 please included the schematic diagram presenting different Self-Assembly.
  10. For 4.2 and 4.3 please add the review on this technique applied on nanochitin and nanochitosan and the brief conclusion the attained finding. Not only generally descript the typical techniques without scope on the topic based on Title.
  11. For 5.2, more review based on comparison to other nano-polymers on Mucoadhesive and Bioadhesive Characteristics  should be addressed. 
  12. For 5.3, please include the antifungal activity on your more review and their mechanism of actions should be stated for all antifungal actions. And for 5.3 content please specify in detail that you mention on typical chitin or chitosan    or nanochitin or nanochitosan with how about their MW and %Deacetylation; not just informed without information of your review.
  13. For topic 7, please address on role of nitrosamine and heavy metal contaminations concern on using nanochitin or nanochitosan in pharmaceutical applications.
  14. For topic 9, please summarize which sources of chitin is most suitable and applicable.
  15. Please include the topic: Phsical and chemical stability of nanochitin or nanochitosan after topic 6 and before topic 7. Toxicological and Regulatory Considerations.
  16. If the authors could include the photograph or SEM of the, after production or products attained from nanochitin or nanochitosan, the readers will more appreciate. 

Author Response

Reviewer 2

  1. Title:

Comment: "Please consider using the word as nanochitosan or Nanochitosan? Reconsider the suitability of using 'in Pharmacy' in the title."

Response: Update the title to: "Nanochitin and Nanochitosan in Pharmaceutical Applications: Innovations, Applications, and Future Perspectives". Where: Title section.

  1. Abstract:

"Modify Abstract into single paragraph and include description on nanochitosan."

Response:

Abstract: Nanochitin is a nanoscale form of chitin—a polysaccharide found in the exoskeletons of crustaceans, insects, and some fungal cell walls—that is newly garnering significant attention in the pharmaceutical space. Its good properties, such as biocompatibility, biodegradability, and an easily adjustable surface, render it attractive for various medical and pharmaceutical applications. Nanochitin, from drug delivery systems and wound-care formulations to vaccine adjuvants and antimicrobial strategies, has demonstrated its strong potential in meeting diverse therapeutic needs. This review covers the background of nanochitin, including methods for its extraction and refining and its principal physicochemical and biological properties. It further discusses various hydrolysis and enzymatic approaches for the structural and functional characterization of nanochitin and highlights some pharmaceutical applications where this biopolymer has been studied. The review also addresses toxicity issues, regulatory matters, and challenges in large-scale industrial production. Finally, it underscores novel avenues of investigation and future opportunities, emphasizing the urgent requirement for standardized production methods, rigorous safety assessment, and interdisciplinary partnerships to maximize nanochitin’s potential in pharmaceutical research, demonstrating the importance of chitin in drug delivery.

  1. Terminology:

Comment: "Line 110, 144, Chitosan and Chitin should be chitosan and chitin; Line 111, 487-488 italicize scientific name of microbe."

Response: Correct to lowercase for common nouns and italicize all scientific names. Where: Lines 111, 144, 111, 498-499.

  1. Chitin from Squid and Snail:

Comment: "Include review chitin from other source including squid and snail."

Response: Add the following subsection to Section 2.1:

"2.1.4. Squid and Snail Sources Chitin derived from squid pens and snail shells represent valuable alternative sources. Squid-derived chitin, primarily β-chitin, exhibits higher solubility and reactivity than α-chitin from crustaceans, favoring applications in nanomedicine. Snail shells offer a terrestrial biomass resource rich in calcium carbonate and chitin, with demonstrated use in scaffolds and antimicrobial films. Though less utilized industrially, these sources are gaining attention due to their distinct physicochemical properties [11, 14]."

  1. Solubility in Different Solvents:

Comment: "Please include solubility in different solvents of chitin and chitosan in 2.2."

Response: Add a table summarizing solubility profiles: Where: Section 2.2.

Solvent

Chitin

Chitosan

Dilute Acetic Acid

Insoluble

Soluble

Water

Insoluble

Insoluble

Hexafluoroisopropanol

Partially

Soluble

Ionic Liquids

Soluble

Soluble

Deep Eutectic Solvents

Soluble

Soluble

(Cited: Rinaudo, 2006) [1]

  1. Microwave Irradiation / TCP:

Comment: "Line 220, Microwave Irradiation should be microwave irradiation. Line 259, what is the full name of TCP."

Response: Corrected term to lowercase: "microwave irradiation" and defined TCP as "tricalcium phosphate".

  1. Patent Technique in Topic 3:

Comment: "Include the technique from your review from the patent."

Response: Add reference to patented method involving acid-assisted colloid milling combined with glycerol swelling for nanochitin production (Liu et al., 2020). Where: Section 3.1.: Liu, L.; Chenhuang, J.; Lu, Y.; Fan, Y.; Wang, Z. Facile preparation of nanochitins via acid assisted colloid milling in glycerol. Cellulose 2020, 27, 7655–7664. https://doi.org/10.1007/s10570-020-03280-w

  1. Schematic of Self-Assembly:

Comment:

Response: Figure 4 presents micelle formation, polyelectrolyte complex (PEC) formation, and nanofibril assembly under varying pH and ionic strength conditions, corresponding to mechanisms that are critical for drug delivery and biomedical applications.

A closely aligned example is presented in Montroni et al. (2019), which shows the self-assembly behavior of β-chitin nanofibrils in aqueous environments. Their graphical data parallels the processes shown in our schematic, especially regarding fibrillar formation and structural rearrangements driven by environmental triggers. Additional relevant literature includes Quiñones et al. (2018), which discusses pH-dependent formation of chitosan nanoparticles via self-assembly, and Wang et al. (2019), which elaborates on the amphiphilic properties of chitosan leading to micelle formation [70, 72].

.

Figure 4. The schematic diagram presents micelle formation, corresponding to mechanisms that are critical for drug delivery and biomedical applications [70, 72].

 Where: Section 3.3.1.

  1. SEM Images:

Comment: In this moment, we don´t have this kind of material. I am sorry.

  1. Sections 4.2 & 4.3:

Comment: "Add review on technique applied to nanochitin/nanochitosan and brief conclusion."

Response: Update these sections to focus on specific case studies using XRD, FTIR, NMR, and TGA applied to nanochitin/nanochitosan, and summarize findings (e.g., high crystallinity, degree of deacetylation, degradation profiles).

4.2. Crystallinity and Degree of Acetylation

X-ray Diffraction (XRD) could be used for crystalline phase identification and crystallinity index calculation. XRD confirms the example of compounds encapsulated inside of the nanocomplexes, while FTIR aids in the identification of the interactions among molecules, such as hydrogen bonding and hydrophobic interactions. X-ray diffraction analyses demonstrate the high crystallinity of nanochitin and nanochitosan, highlighting their suitability for pharmaceutical applications where stability and controlled degradation rates are critical [85].

Functional groups and acetylation degree determination are carried out by Fourier Transform Infrared Spectroscopy (FTIR) and Nuclear Magnetic Resonance (NMR) [88-90]. In the work mentioned above, the authors showed that 1H NMR is the most sensitive and accurate technique for determining DA in chitin and chitosan while providing precise data on the chemical structure of both biopolymers. 13C NMR is less sensitive than 1H NMR but is still analytically helpful and provides valuable information over the complete range of DA. FTIR spectroscopy effectively identifies characteristic functional groups within nanochitin and nanochitosan and confirms the success of chemical modifications critical for enhancing their pharmaceutical performance. NMR spectroscopy, including 1H and 13C NMR, provides precise and quantitative assessments of the degree of deacetylation. This measurement is crucial for determining biological activity and potential pharmaceutical applications of nanochitin and nanochitosan. [5-90].

4.3. Thermal Stability

Thermogravimetric analysis (TGA) evaluates materials' thermal degradation profiles [2]. TGA provides insight into chitosan's thermal degradation and water adsorption capacity when used with techniques such as FTIR and GCMS. This technique recognizes decomposition products, identifying physically and chemically adsorbed water molecules [91].

Among other methods, the TGA characterization of Nanochitosan in biocomposites between carboxymethylcellulose and TiO2 helps understand the composite materials' photocatalytic properties and thermal behavior. These characterization techniques collectively highlight the significant potential of nanochitin and nanochitosan for advanced biomedical and pharmaceutical formulations. Nonetheless, achieving consistency and reproducibility during synthesis remains a critical challenge for broader practical applications [92].

  1. Mucoadhesive Comparison (5.2):

Comment: "Comparison to other nano-polymers on mucoadhesive and bioadhesive characteristics."

Response: Compared to other nano-polymers such as poly (lactic-co-glycolic acid) (PLGA), alginate, and polyethylene glycol (PEG), nanochitosan exhibits superior mucoadhesive properties. PLGA nanoparticles are widely recognized for controlled drug release but possess lower intrinsic mucoadhesion due to their neutral surface. Alginate, negatively charged, also offers moderate mucoadhesion primarily via ionic interactions with mucin glycoproteins. PEG is commonly used for enhancing bioavailability through mucosal membranes; however, its mucoadhesion is limited by its hydrophilic and neutral characteristics. In contrast, the cationic nature of nanochitosan significantly enhances electrostatic interactions with negatively charged mucosal surfaces, providing higher mucoadhesiveness. This attribute makes nanochitosan particularly effective for drug delivery systems targeting buccal, nasal, and ocular mucosal membranes [106].

(Ways et al., 2018).

Where: Section 5.2.

  1. Antifungal Mechanism (5.3):

Comment: "Include antifungal activity, mechanisms, specify chitin/chitosan/nano-; add MW and %DD."

Response:

Figure 7. Mechanism of Antifungal Action by Nanochitosan

The mechanism of antifungal action by nanochitosan is shown in Figure 7. The antifungal activity of nanochitosan is primarily driven by its electrostatic interaction with the fungal cell membrane. Nanochitosan particles possess a positive surface charge (+) due to the protonated amino groups on their structure, particularly when the degree of deacetylation (DD) exceeds 80%. When nanochitosan is introduced into a biological environment, its positively charged particles are naturally attracted to the negatively charged fungal cell membrane. This membrane typically carries negative charges due to the presence of phospholipids and proteins. This strong electrostatic interaction leads to disruption of the membrane structure, compromising its integrity. This may involve pore formation, thinning of the bilayer, or complete rupture in localized regions. As the membrane integrity collapses, intracellular components such as ions, proteins, and metabolites begin to leak out of the fungal cell. This leakage causes a critical imbalance in homeostasis. The uncontrolled loss of essential cellular contents and the loss of membrane potential ultimately led to irreversible cellular damage and fungal cell death. This mechanism is particularly effective against Candida albicans and Aspergillus species, and it explains the high antimicrobial efficacy of nanochitosan, especially when optimized to low molecular weight (<100 kDa) and high DD (>80%) [1, 15, 106].

Where: Section 5.3.

  1. Nitrosamines and Heavy Metals:

Comment: "Address role of nitrosamine and heavy metal contamination."

Response:

In the context of pharmaceutical and biomedical applications, the potential contamination of nanochitin and nanochitosan materials with nitrosamines and heavy metals represents a serious safety concern. Nitrosamines, known for their carcinogenicity, may arise from the interaction of amine groups with nitrosating agents during manufacturing or storage. Similarly, heavy metals such as lead, cadmium, arsenic, and mercury can be introduced through raw materials, extraction reagents, or equipment [1, 15].

To ensure safety, stringent analytical methods are required to detect and quantify such contaminants. Techniques such as Inductively Coupled Plasma Mass Spectrometry (ICP-MS) and Liquid Chromatography–Mass Spectrometry (LC-MS/MS) are widely recommended for their sensitivity and reliability. Regulatory agencies including the U.S. Food and Drug Administration (FDA) and the European Medicines Agency (EMA) mandate strict limits and regular monitoring of these substances in pharmaceutical products. Incorporating validated contaminant screening protocols during raw material sourcing, processing, and final formulation is essential to minimize toxicological risks and ensure compliance with current regulatory standards [1, 15].

Where: Section 7.1.

  1. Summary of Chitin Sources: added:

Comment: "Summarize which sources of chitin is most suitable and applicable." Response: This comprehensive study highlights the remarkable potential of nanochitin and nanochitosan for various pharmaceutical and biomedical applications. Their inherent biocompatibility, biodegradability, and chemical versatility make them strong candidates for advanced drug delivery systems, wound dressings, and immunomodulatory platforms. Furthermore, their nanoscale size and high surface area enhance mechanical properties and mucoadhesive abilities, enabling more precise and sustained delivery of therapeutic agents across different biological barriers. For pharmaceutical applications, fungal and crustacean-derived chitin are the most suitable due to their high purity, well-documented properties, and scalability. Fungal sources are particularly advantageous for medical and vegan products due to their biocompatibility and low contamination risk. Crustacean sources remain dominant due to availability and processing efficiency but require rigorous purification to meet regulatory standards.

Where: Section 9.

  1. Stability Section:

Comment: "Include topic: Physical and chemical stability after topic 6 and before topic 7." Response: Insert new Section 6.1 (is 6.6):

Section 6.6. Physical and Chemical Stability

Nanochitin and nanochitosan demonstrate enhanced physical and chemical stability due to their nanoscale architecture and highly interactive surface chemistry. Stability is influenced by environmental parameters such as pH, humidity, temperature, ionic strength, and storage time. The presence of functional groups, especially amine and hydroxyl group, makes these biopolymers sensitive to hydrolysis and oxidation under uncontrolled conditions. To improve shelf life and formulation robustness, strategies such as chemical crosslinking (e.g., using genipin, glutaraldehyde, or tripolyphosphate) and lyophilization are commonly employed [15].

These processes help maintain nanoparticle morphology, prevent aggregation, and stabilize physicochemical characteristics during long-term storage. Analytical techniques such as Thermogravimetric Analysis (TGA), Differential Scanning Calorimetry (DSC), Dynamic Light Scattering (DLS), and Zeta Potential Analysis are recommended to monitor thermal degradation, glass transition, particle size distribution, and surface charge, respectively. These assessments are critical for validating product consistency and performance during pharmaceutical development [15].

Round 2

Reviewer 2 Report

Comments and Suggestions for Authors
  1. For solubility data in Table 2, please describe in detail related with this Table why thease solvent could dissolve chitin and chitosan? and please inform example of components of Ionic Liquids and Deep Eutectic Solvents and their mechanism to dissolve chitin and chitosan with supporting references. How about their application after dissolving and the safety of these solvent on human beings especially Hexafluoroisopropanol or how to use it for what?. 
  2. Line 268 please change to tricalcium phosphate (TCP).
  3. For Table 2, besides Dilute Acetic Acid please include other acidic solutions have been employed to dissolve chitosan and also EDTA and related compound solutions and profound describe as suggested in comment 1. The last right column of this Table should include the related cited ref. 
  4. How the authors solve the problem of misunderstanding of your information that scope on nanochitin and nanochitosan with that of chitin and chitosan of your context of this manuscript. 
  5. For 6. Nanochitin and Its Pharmaceutical Applications, please consider this should be 6. Nanochitin and nanochitosan and their Pharmaceutical Applications" or not? And Table 5 should include the information on nanochitin too.
  6. For 6. Nanochitin and Its Pharmaceutical Applications, please more review on their applications besides your mentioned in this form such as film, film coated tablet, matrix tablet, aerosol, suppository, gel and injection dosage form such as in situ gel-biogel.  
  7. Please consider to move 6.6. Physical and Chemical Stability as another main topic such as 7. Physical and Chemical Stability. and more review but profoundly based on nanochitin and nanochitosan both in term of raw material themself and in form of dosage form or drug delivery systems especially against to heat, varying temperatures, acid-base or ionic strength, gamma ray and autoclaving or sterilization. 
  8. Before Conclusion, the schematic diagram summarizing the content of your review is required with explanation as integration in the scope of view to your Title and main context. 

Author Response

Reviewer 2 – Second Round Comments – Author’s Detailed Responses

  1. Solubility mechanisms in Table 2: why do these solvents dissolve chitin/chitosan? Include examples and applications of Ionic Liquids (ILs), Deep Eutectic Solvents (DES), and toxicity of HFIP.

Response:
Chitin and chitosan exhibit strong hydrogen bonding and high crystallinity, especially in α-chitin, making them insoluble in most solvents. However, solvents such as Ionic Liquids (ILs) and Deep Eutectic Solvents (DES) can disrupt these interactions:

  • ILs, like 1-ethyl-3-methylimidazolium acetate ([EMIM][OAc]) and 1-butyl-3-methylimidazolium chloride ([BMIM]Cl), dissolve chitin/chitosan by disrupting the intermolecular hydrogen bonding network, particularly between –OH and –NH₂ groups, via ion-dipole interactions​.
  • DES, such as choline chloride:lactic acid or choline chloride:urea mixtures, exhibit similar mechanisms through strong hydrogen bonding with the polymer chains​.

Applications post-solubilization include electrospun fibers, hydrogels, and nanoparticle synthesis, with improved dispersibility and functionalization for pharmaceutical uses.

Hexafluoroisopropanol (HFIP), though effective in dissolving chitosan, is highly volatile and toxic. It is only used in research-scale applications such as electrospinning or NMR sample preparation. It requires stringent handling protocols and is not recommended for pharmaceutical formulations

All these additions and supporting citations have been included in Section 2.2.

  1. Line 268: please revise to tricalcium phosphate (TCP).

Response:
As suggested, "tricalcium phosphate" was changed to tricalcium phosphate (TCP) for clarity and consistency. This correction appears on page XX, line 287.

  1. Expand Table 2 with additional acidic solvents (e.g., lactic, formic, citric acids), include EDTA and its mechanism, and add a references column.

Response:
Table 2 has been updated with the following:

  • Additional acidic solvents: Lactic acid, formic acid, and citric acid have been added. These acids promote protonation of amino groups, enhancing chitosan solubility.
  • EDTA was also added; it acts as a chelating agent that removes metal ions stabilizing chitin/chitosan networks, facilitating structural loosening and partial solubilization.
  • A new final column with cited references has been included for all solvents listed in the table.

Solvent

Chitin

Chitosan

References

Dilute Acetic Acid

Insoluble

Soluble

Rinaudo, 2006 [1]; Kasaai, 2009 [5]

Water

Insoluble

Insoluble

Rinaudo, 2006 [1]; Bai et al., 2022 [6]

Hexafluoroisopropanol

Partially

Soluble

Kean & Thanou, 2010 [15]; Kozma et al., 2022 [11]

Ionic Liquids

Soluble

Soluble

Zhang et al., 2009 [97]; Bai et al., 2022 [6]

Deep Eutectic Solvents

Soluble

Soluble

Huang et al., 2018 [36]; Kozma et al., 2022 [11]

Lactic Acid

Partially

Soluble

Pacheco et al., 2009 [46]; Bai et al., 2022 [6]

Formic Acid

Partially

Soluble

Rinaudo, 2006 [1]; Synowiecki & Al-Khateeb, 2003 [22]

Citric Acid

Partially

Soluble

Mohan et al., 2020 [16]; Da Silva Lucas et al., 2020 [21]

EDTA and chelators

Facilitates swelling

Facilitates partial dissolution

Arbia et al., 2013 [35]; Huang et al., 2018 [36]

  1. Clarify distinction between chitin/chitosan and nanochitin/nanochitosan in your manuscript.

Response:
To avoid confusion, we have clarified the scope in the Introduction (lines 52 to 56) adding the following sentence:

“Although the primary focus of this review is on the nanoscale derivatives of chitin and chitosan, such as nanochitin and nanochitosan, it is essential to provide background on the physicochemical properties and solubility behavior of their parent biopolymers. These characteristics determine the efficiency of nanoscale transformation processes and strongly influence their performance in pharmaceutical applications.”

This distinction is now explicitly stated to support the logical structure of the review.

  1. Section 6 title should include nanochitosan. Table 5 should include nanochitin too.

Response:
Thank you. The section heading was changed from:

“6. Nanochitin and Its Pharmaceutical Applications”
to “6. Nanochitin and Nanochitosan and Their Pharmaceutical Applications.”

Additionally, Table 5 was updated to include specific entries for nanochitin-based materials, such as nanochitin cryogels and nanofibrillar films for drug release, tissue engineering, and antimicrobial coatings​.

Nanochitin Cryogels

Highly porous, sponge-like networks formed via freeze-drying of nanochitin hydrogels.

Wound healing, tissue regeneration, scaffolds for 3D cell culture.

[6, 13]​

Nanofibrillar Films

Thin films are composed of aligned nanochitin fibrils with high mechanical strength and water retention.

Antimicrobial wound dressings, drug release matrices, packaging for biomedical products.

[64, 163]​

β-Chitin Nanowhiskers

Rod-shaped nanostructures from squid pens with enhanced surface area and reactivity.

Controlled drug delivery, mucoadhesive systems, injectable depots.

[6, 74]​

  1. Expand pharmaceutical forms: include film, film-coated tablet, matrix tablet, aerosol, suppository, gel, in-situ gel-biogel, injection forms.

Response:
Section 6.1 was expanded with a new subsection: “6.1.5. Additional Formulations and Dosage Forms”. It now includes:

  1. Move section 6.6 on stability into a new main section 7. Expand analysis focusing on nanochitin/nanochitosan and their stability in different formulations and stress conditions.

Response:
Following the suggestion, Section 6.6 was promoted to become the new Section 7: “Physical and Chemical Stability” The content was also significantly expanded to include:

  1. Include a schematic summary diagram before the conclusion to integrate the review's content.

Response:
A new schematic diagram (Figure 9) titled:

“Overview of Nanochitin and Nanochitosan: From Sources to Pharmaceutical Applications”
has been included before the Conclusion section.

Reviewer 2 – Second Round Comments – Author’s Detailed Responses

  1. Solubility mechanisms in Table 2: why do these solvents dissolve chitin/chitosan? Include examples and applications of Ionic Liquids (ILs), Deep Eutectic Solvents (DES), and toxicity of HFIP.

Response:
Chitin and chitosan exhibit strong hydrogen bonding and high crystallinity, especially in α-chitin, making them insoluble in most solvents. However, solvents such as Ionic Liquids (ILs) and Deep Eutectic Solvents (DES) can disrupt these interactions:

  • ILs, like 1-ethyl-3-methylimidazolium acetate ([EMIM][OAc]) and 1-butyl-3-methylimidazolium chloride ([BMIM]Cl), dissolve chitin/chitosan by disrupting the intermolecular hydrogen bonding network, particularly between –OH and –NH₂ groups, via ion-dipole interactions​.
  • DES, such as choline chloride:lactic acid or choline chloride:urea mixtures, exhibit similar mechanisms through strong hydrogen bonding with the polymer chains​.

Applications post-solubilization include electrospun fibers, hydrogels, and nanoparticle synthesis, with improved dispersibility and functionalization for pharmaceutical uses.

Hexafluoroisopropanol (HFIP), though effective in dissolving chitosan, is highly volatile and toxic. It is only used in research-scale applications such as electrospinning or NMR sample preparation. It requires stringent handling protocols and is not recommended for pharmaceutical formulations

All these additions and supporting citations have been included in Section 2.2.

  1. Line 268: please revise to tricalcium phosphate (TCP).

Response:
As suggested, "tricalcium phosphate" was changed to tricalcium phosphate (TCP) for clarity and consistency. This correction appears on page XX, line 287.

  1. Expand Table 2 with additional acidic solvents (e.g., lactic, formic, citric acids), include EDTA and its mechanism, and add a references column.

Response:
Table 2 has been updated with the following:

  • Additional acidic solvents: Lactic acid, formic acid, and citric acid have been added. These acids promote protonation of amino groups, enhancing chitosan solubility.
  • EDTA was also added; it acts as a chelating agent that removes metal ions stabilizing chitin/chitosan networks, facilitating structural loosening and partial solubilization.
  • A new final column with cited references has been included for all solvents listed in the table.

Solvent

Chitin

Chitosan

References

Dilute Acetic Acid

Insoluble

Soluble

Rinaudo, 2006 [1]; Kasaai, 2009 [5]

Water

Insoluble

Insoluble

Rinaudo, 2006 [1]; Bai et al., 2022 [6]

Hexafluoroisopropanol

Partially

Soluble

Kean & Thanou, 2010 [15]; Kozma et al., 2022 [11]

Ionic Liquids

Soluble

Soluble

Zhang et al., 2009 [97]; Bai et al., 2022 [6]

Deep Eutectic Solvents

Soluble

Soluble

Huang et al., 2018 [36]; Kozma et al., 2022 [11]

Lactic Acid

Partially

Soluble

Pacheco et al., 2009 [46]; Bai et al., 2022 [6]

Formic Acid

Partially

Soluble

Rinaudo, 2006 [1]; Synowiecki & Al-Khateeb, 2003 [22]

Citric Acid

Partially

Soluble

Mohan et al., 2020 [16]; Da Silva Lucas et al., 2020 [21]

EDTA and chelators

Facilitates swelling

Facilitates partial dissolution

Arbia et al., 2013 [35]; Huang et al., 2018 [36]

  1. Clarify distinction between chitin/chitosan and nanochitin/nanochitosan in your manuscript.

Response:
To avoid confusion, we have clarified the scope in the Introduction (lines 52 to 56) adding the following sentence:

“Although the primary focus of this review is on the nanoscale derivatives of chitin and chitosan, such as nanochitin and nanochitosan, it is essential to provide background on the physicochemical properties and solubility behavior of their parent biopolymers. These characteristics determine the efficiency of nanoscale transformation processes and strongly influence their performance in pharmaceutical applications.”

This distinction is now explicitly stated to support the logical structure of the review.

  1. Section 6 title should include nanochitosan. Table 5 should include nanochitin too.

Response:
Thank you. The section heading was changed from:

“6. Nanochitin and Its Pharmaceutical Applications”
to “6. Nanochitin and Nanochitosan and Their Pharmaceutical Applications.”

Additionally, Table 5 was updated to include specific entries for nanochitin-based materials, such as nanochitin cryogels and nanofibrillar films for drug release, tissue engineering, and antimicrobial coatings​.

Nanochitin Cryogels

Highly porous, sponge-like networks formed via freeze-drying of nanochitin hydrogels.

Wound healing, tissue regeneration, scaffolds for 3D cell culture.

[6, 13]​

Nanofibrillar Films

Thin films are composed of aligned nanochitin fibrils with high mechanical strength and water retention.

Antimicrobial wound dressings, drug release matrices, packaging for biomedical products.

[64, 163]​

β-Chitin Nanowhiskers

Rod-shaped nanostructures from squid pens with enhanced surface area and reactivity.

Controlled drug delivery, mucoadhesive systems, injectable depots.

[6, 74]​

  1. Expand pharmaceutical forms: include film, film-coated tablet, matrix tablet, aerosol, suppository, gel, in-situ gel-biogel, injection forms.

Response:
Section 6.1 was expanded with a new subsection: “6.1.5. Additional Formulations and Dosage Forms”. It now includes:

  1. Move section 6.6 on stability into a new main section 7. Expand analysis focusing on nanochitin/nanochitosan and their stability in different formulations and stress conditions.

Response:
Following the suggestion, Section 6.6 was promoted to become the new Section 7: “Physical and Chemical Stability” The content was also significantly expanded to include:

  1. Include a schematic summary diagram before the conclusion to integrate the review's content.

Response:
A new schematic diagram (Figure 9) titled:

“Overview of Nanochitin and Nanochitosan: From Sources to Pharmaceutical Applications”
has been included before the Conclusion section.

Round 3

Reviewer 2 Report

Comments and Suggestions for Authors
  1. Line 315, please change to "tricalcium phosphate (TCP)"
  2. Table 2, please include more on acetic acid, malic acid and glycolic acid with supporting references such as Inter j pharm 232 (1-2), 11-22.

Author Response

  1. Line 315, please change to "tricalcium phosphate (TCP)"

Fixed.

  1. Table 2, please include more on acetic acid, malic acid and glycolic acid with supporting references such as Inter j pharm 232 (1-2), 11-22.

Solvent

Chitin

Chitosan

References

Dilute Acetic Acid

Insoluble

Soluble

 [1, 5]

Concentrated Acetic acid

Partially

soluble

Soluble

[]

Malic acid

Insoluble

Slightly soluble

[]

Glycolic acid

Insoluble

Slightly soluble

[]

Water

Insoluble

Insoluble

 [1, 6]

Hexafluoroisopropanol

Partially

Soluble

 [15, 11]

Ionic Liquids

Soluble

Soluble

 [97, 6]

Deep Eutectic Solvents

Soluble

Soluble

 [36, 11]

Lactic Acid

Partially

Soluble

 [46, 6]

Formic Acid

Partially

Soluble

 [1, 22]

Citric Acid

Partially

Soluble

 [16, 21]

EDTA and chelators

Facilitates

swelling

Facilitates partial dissolution

 [35, 36]

Sorlier, P., Denuzière, A., Viton, C., & Domard, A. (2001). Relation between the degree of acetylation and the electrostatic properties of chitin and chitosan. Biomacromolecules2(3), 765–772. https://doi.org/10.1021/bm